# Zero-Shot Blind-Spot Image Denoising via Cross-Scale Non-Local Pixel Refilling

**Qilong Guo    Tianjing Zhang    Zhiyuan Ma    Hui Ji**
Department of Mathematics, National University of Singapore, 119076, Singapore
{qilong.guo, tianjingzhang, e0983565}@u.nus.edu, matjh@nus.edu.sg

## Abstract

Blind-spot denoising (BSD) method is a powerful paradigm for zero-shot image denoising by training models to predict masked target pixels from their neighbors. However, they struggle with real-world noise exhibiting strong local correlations, where efforts to suppress noise correlation often weaken pixel-value dependencies, adversely affecting denoising performance. This paper presents a theoretical analysis quantifying the impact of replacing masked pixels with observations exhibiting weaker noise correlation but potentially reduced similarity, revealing a trade-off that impacts the statistical risk of the estimation. Guided by this insight, we propose a computational scheme that replaces masked pixels with distant ones of similar appearance and lower noise correlation. This strategy improves the prediction by balancing noise suppression and structural consistency. Experiments confirm the effectiveness of our method, outperforming existing zero-shot BSD methods.

## 1 Introduction

Image denoising, which aims to restore clean images from their noisy observations, is a basic problem in image processing. Deep learning, particularly supervised methods, has become a popular approach, achieving impressive performance on synthetic noise [1, 2, 3, 4, 5, 6]. These models are trained using synthetic noise, such as additive white Gaussian noise (AWGN). Consequently, their generalization to real-world noise is limited [7, 8], as such noise is often spatially correlated. To address this domain gap, several real-world datasets have been constructed, such as SIDD [9] and FMDD [10]. However, these datasets cover only a limited range of the diverse noise patterns encountered in practice. Alternative approaches, including the use of unpaired noisy-clean image collections [11, 12, 13] or multiple noisy observations [14, 15], still rely on clean images, which are challenging to obtain in many domains, such as scientific and medical imaging.

Self-supervised image denoising, sometimes also referred to as unsupervised denoising, enables training directly from noisy observations, bypassing the need for clean images. One common strategy constructs pseudo pairs to define self-supervised losses [16, 17, 18]. Another popular strategy is the BSD method, which trains neural networks (NNs) to predict each target pixel from its neighboring pixels, excluding itself [19, 20, 21, 22, 23, 24, 25, 26]. These methods show promising performance on real-world images. However, they depend on large external noisy datasets, which may be unavailable or misaligned with the target domain. This limitation has motivated zero-shot denoising methods, which avoid external data by adapting a model to each noisy image at test time.

**Zero-shot denoising at testing time:** Zero-shot methods adapt an untrained model directly to a noisy image at test time. The pioneering Deep Image Prior (DIP) [27] demonstrates that convolutional neural networks (CNNs) inherently favor structured patterns over noise, enabling zero-shot denoising

via early stopping. The blind-spot approach is also effective in this setting, training a model from scratch at test time to predict masked pixels from their unmasked neighbors [28, 29, 30]. Blind-spot methods have shown strong performance in the setting of zero-shot denoising.

**Overview of BSD paradigm:** The practical relevance of zero-shot denoising and the effectiveness of BSD techniques motivate our investigation into further refinement of BSD to meet the demand for a powerful zero-shot solution for denoising real-world images.

Let $y = x + n$ be a noisy image, where $x$ denotes truth image and $n$ is noise. For a target pixel $x_i$ with noisy observation $y_i$, let $\Omega_i$ denote the index set of the pixels within its receptive field. In supervised learning, a denoising NN $\mathcal{D}_\phi : y \to x$ is trained to minimize the following loss:

$$\sum_i |\mathcal{D}_\phi(\{y_j\}_{j \in \Omega_i}) - x_i|^2 \quad \text{(Supervised)}, \tag{1}$$

which measures the discrepancy between the prediction and truth for all pixels.

In zero-shot denoising, as truth $x_i$ is unknown, BSD methods train the NN to approximate the loss (1). There are two BSD implementations:

- *Architecture-based BSD*: The NN is designed to enforce a blind spot so that the receptive field never contains the target pixel [31, 22, 23, 24, 25, 26, 32].

- *Mask-based BSD*: A standard backbone is kept, but a random mask removes selected pixels from the input and the loss is computed only on those masked pixels [20, 19, 28, 21, 29, 30].

Overall, mask-based BSD is more appealing than architecture-based BSD in practical usage. It imposes no architectural changes: A simple input mask and modified loss suffice, so any off-the-shelf NN model can be used. Moreover, mask ratio can be flexibly tuned. Specifically, let $M \in \{0, 1\}^{H \times W}$ be a Bernoulli mask of probability $p$. Let $\mathcal{M}_0 := \{i : m_i = 0\}$ be the index set of masked pixels and $\mathcal{M}_1 := \{i : m_i = 1\}$ the index set of unmasked pixels. Then, its loss can be formed as:

$$\sum_{i \in \mathcal{M}_0} |\mathcal{D}_\phi(\{y_j\}_{j \in \Omega_i \cap \mathcal{M}_1}) - y_i|^2 \quad \text{(BSD)}. \tag{2}$$

**Key challenges in BSD:** To maximize the effectiveness of BSD, the mask-based loss (2) should be a good approximation to the supervised loss (1). By comparing the two losses, we can see that the BSD loss (2) is a good approximation to the supervised loss (1) if the following conditions are satisfied:

- *Mask Ratio:* The ratio of masked pixels in image, $r = \frac{|\mathcal{M}_0|}{|\mathcal{M}_0| + |\mathcal{M}_1|}$ must be large enough; otherwise, the loss (2) involves too few pixels, leading to high variance. Conversely, $r$ must not be too large, as it will reduce the number of unmasked pixels in $\mathcal{M}_1 \cap \Omega_i$ for prediction.

- *Noise Correlation:* The noise of the unmasked pixels in $\mathcal{M}_1 \cap \Omega_i$ should be nearly independent to that of the masked target pixels $y_i$; otherwise, the loss (2) will incur a significant bias related to the supervised loss (1).

BSD enables self-supervised learning by predicting certain pixels from their surrounding pixels while excluding themselves, thus preventing over-fitting to their noisy counterparts. However, as shown in the discussion above, the performance of BSD faces two key challenges:

1. *Local correlation of real-world noise*: Real-world noise shows strong local spatial correlation. The noise of nearby pixels $\{y_j\}_{j \in \mathcal{M}_1 \cap \Omega_i}$ is then correlated with that of the target pixel $x_i$, violating the independence assumption in BSD and introducing bias to the loss function (2).

2. *Insufficient related pixels for prediction*: As the ratio $r$ needs to be sufficiently large, the number of unmasked pixels in $\mathcal{M}_1 \cap \Omega_i$ used to predict the target masked pixel $x_i$ is limited. This can lead to insufficient information for an accurate prediction.

Most existing work addresses these challenges by removing more nearby pixels and including more distant ones [22, 24]. However, while exhibiting weak noise correlation, distant pixels also have weaker value correlation with the target pixel, which can hinder accurate prediction.

**Motivations and main idea:** The challenges outlined above, limited predictive pixels due to masking and strong local noise correlation, highlight a fundamental trade-off in BSD, the trade-off between

noise decorrelation and pixel correlation. Existing approaches mitigate these issues by including spatially distant pixels, with lower noise correlation to the masked pixels. However, such pixels are likely less related to the target, reducing value correlation and impairing prediction accuracy. Such a trade-off between noise decorrelation and structural similarity remains a challenge in existing works. Moreover, the reduced predictive context caused by masking is not fully addressed.

Motivated by these observations, we propose to tackle such a challenge by pixel refilling, that is all masked pixels are replaced by other noisy pixels from the same image when being passed to the NN as the input. The question is then how to select the pixels to be used for refilling. In this paper, we analyze the trade-off between noise decorrelation and value correlation for linear denoising NNs.

Briefly, for a masked target pixel $x_0$ with a noisy observation $y_0 = x_0 + n_0$, suppose its receptive field contains a total of $M$ pixels, partitioned into two disjoint subsets:

$$\{y_j = x_j + n_j\}_{j=1}^{pM} \text{ (un-masked pixels)}; \qquad \{\tilde{y}_j = \tilde{x}_j + \tilde{n}_j\}_{j=1}^{(1-p)M} \text{ (re-filled masked pixels)}.$$

The values and noise of both subsets are correlated to $x_0$ and $n_0$ as follows:

$$x_j = (1 - \mu_j)x_0, \; \tilde{x}_j = (1 - \tilde{\mu}_j)x_0; \quad n_j = \lambda_n n_0 + \sqrt{1 - \lambda_n^2}\epsilon_j, \; \tilde{n}_j = \tilde{\lambda}_n n_0 + \sqrt{1 - \tilde{\lambda}_n^2}\tilde{\epsilon}_j,$$

where random variables $\{\mu_j\}_j$, $\{\tilde{\mu}_j\}_j$, $n_0$, $\{\epsilon_j\}_j$ and $\{\tilde{\epsilon}_j\}_j$ are independent with zero mean, variance $\sigma^2$, $\sigma_v^2$ and $\tilde{\sigma}_v^2$ for three noises. Here, $\lambda_n$ measures noise correlation: smaller $\lambda_n$ indicates weaker correlation between $n_j$ and $n_0$, same for $\tilde{\lambda}_n$. $\sigma_v$ and $\tilde{\sigma}_v$ measure the pixel value correlation: Lower $\sigma_v$ and $\tilde{\sigma}_v$ imply stronger correlation between the related predictive pixels and the target $x_0$.

In this paper, we quantify the prediction risk of a linear model under the above assumptions:

$$\mathcal{R}_\phi = \mathbb{E}_{\{\mu_j\}, \{\tilde{\mu}_j\}, \{\epsilon_j\}, \{\tilde{\epsilon}_j\}, n_0}[\|\mathcal{D}_\phi(\{y_j\}_j \cup \{\tilde{y}_j\}) - x_0\|_2^2].$$

From the resulting Bayesian risk analysis for a linear model, we can draw several interesting conclusions which guide the design of our BSD method with pixel refilling. For example,

- The noise correlation term $\tilde{\lambda}_n$ is amplified by $M$, while the value correlation $\tilde{\sigma}_v^2$ is not. Thus, for refilled pixels, reducing noise correlation with the target $n_0$ is more critical than increasing value similarity (i.e., lowering $\tilde{\sigma}_v^2$).

- When the receptive field is sufficiently large ($M$ is large), the prediction risk can still be reduced even if the refilled pixels exhibit lower value correlation than the unmasked ones (i.e., $\tilde{\sigma}_v^2 > \sigma_v^2$), provided they have lower noise correlation with the target pixel ($\tilde{\lambda}_n < \lambda_n$).

These insights suggest it more advantageous to select pixels with lower noise correlation to the target pixel, with a reasonable degree of pixel similarity. To achieve this, one must seek pixels that are spatially or statistically distant yet structurally related to the target. Motivated by this idea, we design a pixel refilling strategy that matches pixels to distant-origin counterparts to reduce noise correlation, with structural similarity ensured by the strong cross-scale self-similarity of natural images, a property widely exploited by classic non-local methods [33, 34] and super-resolution approaches [35, 36, 37].

**Main contributions:** This work addresses two central challenges in BSD: limited predictive pixels due to masking and biased loss caused by local noise correlation. Our contributions are twofold:

- *Theoretical Analysis*: We present a Bayesian-risk analysis of a linear BSD model, revealing a trade-off between noise decorrelation and pixel similarity. It shows replacing masked pixels with others of low noise correlation, despite weaker pixel similarity, can reduce prediction risk.

- *Practical algorithm*: Motivated by this insight, we propose a novel pixel refilling strategy for BSD method that fills masked pixels with carefully selected noisy pixels from the same image across scales, ensuring lower noise correlation and sufficient pixel correlation.

Extensive experiments on real and synthetic noisy images demonstrate the effectiveness of our approach, which in general outperforms existing zero-shot BSD methods.

## 2 Related Work

**Supervised denoising:** Supervised denoising methods learn to directly map noisy images to clean ones on many paired images. Popular architectures include CNN-based models (e.g., DnCNN [1],

complex-valued CNN [38] and NAFNet [39]) and transformer-based models (e.g., Restormer [6] and Swin-IR [5]). However, their performance often degrades on real-world noise, as models trained on synthetic noise do not generalize well. One approach to relax the need for paired data by training on unpaired noisy-clean images [11, 40, 12, 13, 4, 41] or using multiple noisy observations [14, 15], but these still require either clean references or precise alignment, limiting their practicality.

**Self-supervised denoising:** Self-supervised denoising methods train NNs using only noisy images, removing the need for clean references. A dominant paradigm is the blind-spot scheme, which prevents the network from seeing the center pixel during prediction via architectural designs such as occluded receptive fields [42], center-masked convolutions with dilated layers [31], or directional self-attention in transformers [26]. Alternatively, masking-based blind-spot methods like Noise2Void [19] and Noise2Self [20] define losses only over masked pixels to avoid overfitting, with extensions to global-aware mask mapping [21]. While blind-spot schemes perform well for independent noise, their performance decreases for local correlated noise. Pixel-shuffle downsampling (PD) is introduced in [31, 43] to access distant pixels with weaker noise correlation by rearranging pixels from different spatial locations into multiple channels. Although PD reduces noise correlation, it also weakens local value correlation. APBSN [22] refines this by using a large stride during training and a smaller one during testing. LG-BPN [23] extends this with dense sampling and a global branch to improve texture recovery. PUCA [24] uses patch shuffle operations to enhance global context. SS-BSN [25] combines grid self-attention with a simplified D-BSN [31]. TBSN [32] and SelfFormer [26] employ transformer-based blind-spots. Beyond blind-spot models, other self-supervised strategies include recorruption-based methods R2R [16] and its extension [44] to general noise, downsample-based pairing NBR2NBR [17], and cyclic noise decomposition CVF-SID [45].

**Zero-shot denoising:** Zero-shot denoising methods adapt an untrained model to each input at test time, thereby eliminating the need for external training datasets required by self-supervised methods. The pioneer work is DIP [27], which exploits the inductive bias of CNNs to fit structured content faster than noise. Subsequent methods extend this idea: Self2Self [28] combines the blind-spot principle with dropout ensembling; Noise2Fast [46] and ZS-N2N [47] use downsample-based pairing; and Pixel2Pixel [48] creates patch-level matches at the original scale. To better handle real-world noise, Zheng *et al.* [49] project noisy images into a latent AWGN-like space for denoising. ScoreDVI [50] includes score priors into Bayesian inference, TTT-MIM [51] adopts an AWGN prior and adapts to shifted domain, while MASH [29] uses adaptive masking ratio and local shuffling to reduce noise correlation. These methods replace masked pixels with zeros during training, which limits the number of predictive pixels. To address this, BSN-INS [30] employs implicit neural representation (INR) to resample noisy pixels, yielding more predictive pixels with lower noise correlation.

**Image restoration using patch-wise self-similarity:** Patch-wise self-similarity is a well-known phenomenon in natural images, where similar image structures can be found at different locations and scales. This property is first exploited by non-local mean [33] for image denoising, BM3D [34] extends it by grouping similar patches for collaborative filtering. In super-resolution, cross-scale self-similarity has been extensively studied; see, e.g. [35, 36, 37]. However, the self-similarity property has not been fully utilized in BSD for better performance on real-world image denoising.

## 3   Method

We analyze prediction risk in linear BSD models and introduce a mask-based zero-shot BSD method that addresses two challenges: correlated noise and scarce predictive pixels. Our solution is a pixel refilling strategy leveraging natural image self-similarity.

### 3.1   Risk Analysis for Linear BSD Models with Pixel Refilling

We consider a typical mask-based BSD model trained using the loss function (2), with a random mask $M$. Let $x_0$ denote one masked pixel to be predicted. Its receptive field $\Omega_0$ contains $M$ pixels, of which $|\mathcal{M}_0 \cap \Omega_0|$ are masked, leaving only $|\mathcal{M}_1 \cap \Omega_0|$ unmasked pixels available for predicting $x_0$. To alleviate the issue of insufficient predictive information [42], we propose to replace masked pixels with values denoted by $\{\tilde{y}_j\}_{j=1}^{|\mathcal{M}_0 \cap \Omega_0|}$. That is, the input for predicting $x_0$ is

$$\hat{y}_j = m_j y_j + (1 - m_j)\tilde{y}_j, \qquad \text{for } j = 1, \ldots, M, \tag{3}$$

where $m_j$s are independent random samples from Bernoulli($p$), which takes the value 1 with probability $p$ and 0 with probability $1 - p$. Following the same notation as in Sec 1, we assume that the unmasked pixels $\{y_j\}_j$ and the refilled pixels $\{\tilde{y}_j\}_j$ are correlated with the target pixel $x_0$ by

$$y_j = (1 - \mu_j)x_0 + n_j, \qquad \tilde{y}_j = (1 - \tilde{\mu}_j)x_0 + \tilde{n}_j;$$
$$n_j = \lambda_n n_0 + \sqrt{1 - \lambda_n^2}\epsilon_j, \qquad \tilde{n}_j = \tilde{\lambda}_n n_0 + \sqrt{1 - \tilde{\lambda}_n^2}\tilde{\epsilon}_j, \tag{4}$$

where as assumed, all the variables $\{\mu_j\}_j$, $\{\tilde{\mu}_j\}_j$, $n_0$, $\{\epsilon_j\}_j$ and $\{\tilde{\epsilon}_j\}_j$ are i.i.d. random variables with zero mean, where $\sigma_v^2$, $\tilde{\sigma}_v^2$ and $\sigma^2$ are variances for the latter three noises, respectively.

*Remark* 1 (Interpretation of Parameters). The parameters $\lambda_n$, $\tilde{\lambda}_n$, $\sigma_v$, and $\tilde{\sigma}_v$ quantify noise and value correlations for unmasked pixels and refilled pixels to $n_0$ and $x_0$ respectively. Smaller $\lambda_n, \tilde{\lambda}_n$ indicate weaker local noise correlation, while smaller $\sigma_v, \tilde{\sigma}_v$ imply stronger similarity in pixel values.

Consider a linear model $\mathcal{D}_{\boldsymbol{a}}$ parametrized by $\boldsymbol{a} = \{a_j\}_{j=1}^M$: $\mathcal{D}_{\boldsymbol{a}}(\{\hat{y}_j\}_{j=1}^M) = \sum_{j=1}^M a_j \cdot \hat{y}_j$. When using such a linear model to predict $x_0$, the corresponding risk is then defined by:

$$\mathcal{R}_{\boldsymbol{a}} = \mathbb{E}_{\{m_j\},\{\mu_j\},\{\tilde{\mu}_j\},n_0,\{\epsilon_j\},\{\tilde{\epsilon}_j\}}[\|\mathcal{D}_a(\{\hat{y}_j\}_j) - x_0\|_2^2]. \tag{5}$$

Then, a linear model $\mathcal{D}_{\boldsymbol{a}}$ is learned using the following self-supervised loss:

$$\mathcal{D}_{\boldsymbol{a}^*}(\{\hat{y}_j\}_{j=1}^M) = \sum_{j=1}^M a_j \cdot \hat{y}_j, \text{ with } \boldsymbol{a}^* := \operatorname{argmin}_{\boldsymbol{a} \in \mathbb{R}^M}\mathbb{E}[\|\sum_{j=1}^M a_j \cdot \hat{y}_j - y_0\|_2^2]. \tag{6}$$

Then, we have

**Proposition 1.** *The risk of the linear estimator $\mathcal{D}_{\boldsymbol{a}^*}$ defined by* (6) *is:*

$$\mathcal{R}_{\boldsymbol{a}^*} = x_0^2 - \frac{M(x_0^4 - (p\lambda_n + (1-p)\tilde{\lambda}_n)^2\sigma^4)}{Mx_0^2 + (M-1)\sigma^2(p\lambda_n + (1-p)\tilde{\lambda}_n)^2 + p\sigma_v^2 x_0^2 + (1-p)\tilde{\sigma}_v^2 x_0^2 + \sigma^2}. \tag{7}$$

*Proof.* See Appendix B for the details.

It can be seen that the risk $\mathcal{R}_{\boldsymbol{a}^*}$ decreases when the refilled pixels exhibit a lower noise correlation with $n_0$ (smaller $\tilde{\lambda}_n$) and a stronger value correlation with $x_0$ (smaller $\tilde{\sigma}_v^2$). This observation is consistent with intuition. A natural follow-up question is: What if we must trade off between noise and value correlation? Should we prioritize reducing noise correlation, or favor higher value similarity? Suppose $M$ is sufficiently large, then the risk $\mathcal{R}_{\boldsymbol{a}^*}$ in (7) is dominated by the term

$$x_0^2 - \frac{x_0^4 - (p\lambda_n + (1-p)\tilde{\lambda}_n)^2\sigma^4}{x_0^2 + \sigma^2(p\lambda_n + (1-p)\tilde{\lambda}_n)^2} = x_0^2 + \sigma^2 - \frac{x_0^4 + x_0^2\sigma^2}{x_0^2 + \sigma^2(p\lambda_n + (1-p)\tilde{\lambda}_n)^2}. \tag{8}$$

This implies the risk is primarily influenced by the noise correlation term $\tilde{\lambda}_n$ of the refilled pixels, not by the value correlation parameter $\tilde{\sigma}_v^2$. This suggests that, for refilled pixels, reducing their noise correlation (lowering $\tilde{\lambda}_n$) is more important than increasing their value similarity (lowering $\tilde{\sigma}_v^2$).

There are some simple refilling techniques in practice, e.g., randomly selecting neighboring pixels to fill the masked pixels. In such a case, the refilled values $\{\tilde{y}_j\}$ satisfy $\tilde{\lambda}_n = \lambda_n$ and $\tilde{\sigma}_v^2 = \sigma_v^2$.

**Corollary 1.** *When $\{\tilde{y}_j\}$ satisfy $\tilde{\lambda}_n = \lambda_n$ and $\tilde{\sigma}_v^2 = \sigma_v^2$., the risk of $\mathcal{D}_{\hat{\boldsymbol{a}}^*}$ defined by* (6) *is:*

$$\mathcal{R}_{\hat{\boldsymbol{a}}^*} = x_0^2 - \frac{M(x_0^4 - \lambda_n^2\sigma^4)}{Mx_0^2 + (M-1)\sigma^2\lambda_n^2 + \sigma_v^2 x_0^2 + \sigma^2}. \tag{9}$$

*Proof.* See Appendix B for details.

With sufficiently large $M$, two risks (7) and (9) for two refilling schemes roughly satisfy

$$\mathcal{R}_{\boldsymbol{a}^*} \approx (x_0^2 + \sigma^2) - \frac{x_0^4 + x_0^2\sigma^2}{x_0^2 + \sigma^2\underbrace{(p\lambda_n + (1-p)\tilde{\lambda}_n)^2}} \quad \text{v.s.} \quad (x_0^2 + \sigma^2) - \frac{x_0^4 + x_0^2\sigma^2}{x_0^2 + \sigma^2\underbrace{\lambda_n^2}} \approx \mathcal{R}_{\hat{\boldsymbol{a}}^*}. \tag{10}$$

Thus, as long as the refilled pixels exhibit a lower noise correlation than the neighboring pixels, the risk in the linear model can still be reduced, even if their value correlation is weaker. This indicates that simple refilling techniques, such as randomly selecting neighboring pixels, are suboptimal. More understanding of the risk of pixel refilling can be found in Appendix C.

## 3.2 Non-local Pixel Refilling by Cross-scale Patch Matching

Our analysis shows that the benefit from noise decorrelation generally outweighs the drawback of weakened pixel-value correlations. PD fits this principle very well: by reshuffling pixels from different spatial locations, it prioritizes noise decorrelation at the cost of reducing pixel-value correlation. This trade-off explains why PD remains effective, but also reveals its performance bottleneck. To fully exploit its potential, we propose a pixel refilling scheme that constructs an image which compensates the loss of pixel-value correlation caused by PD while further improving noise decorrelation.

Our pixel refilling strategy is inspired by the classic non-local methods (e.g., non-local means [33] and BM3D [34]), which exploit the self-similarity property of natural images to enrich pixel information. Slightly different from those patch-level denoising methods, our approach operates at the pixel level: for each masked pixel, we search for the patch most similar to the one centering at this pixel while ensuring sufficient spatial distance. Then, use the center pixel of the best match for replacement. Such pixels are likely to have high value correlation with the target while maintaining weak noise correlation. To further improve robustness, we leverage cross-scale self-similarity [35, 37] of natural images. Specifically, the patches used for matching are extracted from a down-sampled version of the input image. This cross-scale refilling produces more desired replacements, yielding improved noise independence together with stronger pixel-value correlation.

Since masked pixels are randomly selected across many instances in mask-based BSD, we construct an auxiliary noisy image $\widetilde{y}$ from the original noisy image $y$ to refill the masked pixels. The construction of such an auxiliary image improves computational efficiency, as patch matching only needs to be performed once. For each instance, if a pixel is selected as masked, it is replaced by the corresponding pixel from $\widetilde{y}$ at the same location. The construction of $\widetilde{y}$ is based on the following steps:

1. *Patch extraction at coarse scale:* The noisy image $y$ is down-sampled with average pooling to construct $y\downarrow_k$. Then, $y$ and $y\downarrow_k$ are partitioned into overlapping patches of same size $N \times N$. The patches are extracted with a stride of $N/4$ along the width and height. Let $Y$ and $\widetilde{Y}$ denote the sets of patches from $y$ and $y\downarrow_k$.

2. *Patch matching and replacement:* For each patch $P_i \in Y$, we identify the patch $\widetilde{P}_j \in \widetilde{Y}$ with the smallest $\ell_2$-distance $\|P_i - \widetilde{P}_j\|_2$ among all patches in $\widetilde{Y}$. Then, $P_i$ is replaced with $\widetilde{P}_j$.

3. *Construction of $\tilde{y}$:* After replacement, with a patch size of $N \times N$ and a stride of $N/4$, each pixel in $\tilde{y}$, the auxiliary image used for refilling, is covered by 16 overlapping patches. The final pixel value is obtained by averaging its values across all overlapping patches.

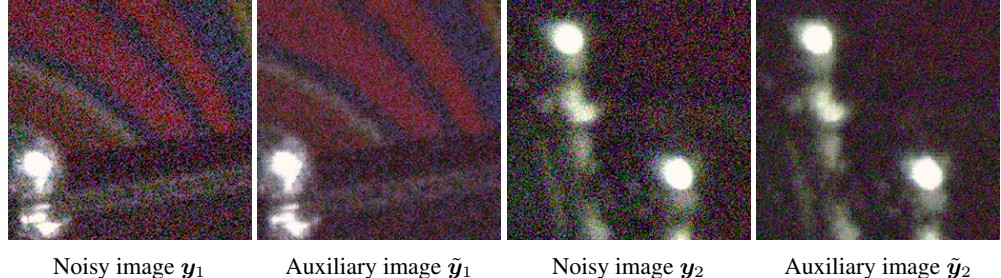

| Noisy image $y_1$ | Auxiliary image $\tilde{y}_1$ | Noisy image $y_2$ | Auxiliary image $\tilde{y}_2$ |

Figure 1: Sample images and their auxiliary images.

The auxiliary image $\tilde{y}$, constructed via the proposed procedure, is used to refill the masked pixels in the original noisy image $y$. Each refilled pixel is selected from a different patch with high similarity to the target. The possible distant origin and down-sampling involved in the patches for matching ensure a lower noise correlation compared to neighboring pixels, while its high similarity to the target pixel $x_0$ suggests strong value correlation. In the implementation, the down-sampling rate $k$ is set to 2, and the down-sampling is generated by applying average pooling on the original noisy image with a window size $2 \times 2$ and a stride 2. See Figure 1 for the illustration of sample images and their auxiliary images. We also provide the illustration for patch matching in Appendix D.

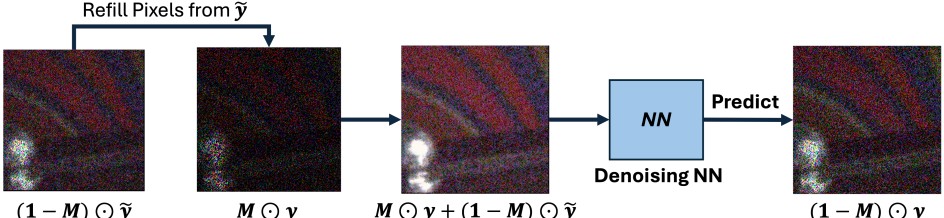

Refill Pixels from $\tilde{y}$

$(1-M) \odot \tilde{y}$      $M \odot y$      $M \odot y + (1-M) \odot \tilde{y}$      $(1-M) \odot y$

Figure 2: Demonstration of BSD with pixel refilling, where $M$ is a random mask. The NN is trained by using both unmasked and refilled pixels to predict masked pixels $(1-M) \odot y$.

### 3.3 BSD with Pixel Refilling and Training

The patch-based strategy first constructs an auxiliary image $\tilde{y}$ by aggregating highly similar but distant patches, thereby reducing local noise correlation while preserving value correlation. Each masked pixel in the original noisy image $y$ is then refilled using the corresponding pixel from $\tilde{y}$. The resulting image, combining refilled and unmasked pixels, is used as input to the denoising network.

Given a random binary mask $M$, we construct the input to the denoising NN $\mathcal{D}_\phi$ by combining unmasked noisy pixels with refilled pixels from the auxiliary image. The NN is trained to predict the masked noisy pixels in a self-supervised manner. Specifically, the BSD-based loss is defined as:

$$\mathcal{L}_{\text{BSN}} = \left\| (1-M) \odot \left( \mathcal{D}_\phi(\hat{y}\downarrow_2^{PD})\uparrow_2^{PD} -y \right) \right\|_1 , \tag{11}$$

where $\hat{y} = M\odot y+(1-M)\odot\tilde{y}$, and $M \sim \text{Bernoulli}(p)$. The parameter $p$ controls the ratio between masked and unmasked pixels. PD with stride 2 symmetrically in both training and testing, denoted as $\downarrow_2^{PD}$ and $\uparrow_2^{PD}$, is implemented in our loss function. To further alleviate possible overfitting caused by the lack of the guidance of truth image during training, we introduce a consistency regularization term [47, 52]:

$$\mathcal{L}_{\text{con}} = \left\| \mathcal{D}_\phi(\hat{y}\downarrow_2^{PD})\uparrow_2^{PD} -\mathcal{D}_\phi(\hat{y}) \right\|_1 , \tag{12}$$

weighted by a scalar $\lambda$. The overall training objective combines the blind-spot reconstruction loss and the consistency regularization term:

$$\mathcal{L} = \mathcal{L}_{\text{BSN}} + \lambda\mathcal{L}_{\text{con}}, \tag{13}$$

where $\lambda$ is a hyperparameter that balances the two terms.

## 4 Experiments

### 4.1 Experiment Setting

**Dataset:** Five datasets, including both real and synthetic ones, are used for benchmarking. (1) **SIDD** [9], which contains real noisy-clean image pairs captured by smartphone cameras under diverse lighting conditions. We use both the validation set (1,280 images of $256 \times 256$) and the benchmark set (1,280 noisy-only images) for evaluation. (2) **DND** [53] consists of 50 pairs of noisy clean images, formed by capturing the same scenes twice with different ISO settings. The high-ISO images serve as noisy inputs, while the corresponding low-ISO images serve as ground-truth references. We use provided $512 \times 512$ images for evaluation on their website. (3) **FMDD** [10], contains real grayscale fluorescence images of biological samples acquired using widefield, confocal, and two-photon microscopes. Image size is $512 \times 512$. (4) **Kodak24**[1] and **McMaster18** [54], two synthetic denoising datasets of color images. We add i.i.d. Gaussian noise with $\sigma = 25, 50$ and follow Pixel2Pixel [48] to crop them into patches of size $256 \times 256$ for evaluation. (5) **fastMRI** [55], a dataset of MRI images. Following TTT-MIM [51], we use its subset, add i.i.d. Gaussian noise with $\sigma = 18$, and crop the images to $224 \times 224$.

**Implementation details:** The proposed model is implemented using PyTorch 2.4.1, CUDA 12.4, and an NVIDIA A6000 GPU. The model is trained using Adam optimizer with an initial learning rate of $10^{-4}$, and other parameters of Adam are set to default. The auxiliary image is constructed with patch

---

[1]`http://r0k.us/graphics/kodak/`

size $N = 8$. Two metrics, peak signal-to-noise ratio (PSNR) and structural similarity index measure (SSIM), are used for quantitative evaluation. Details of the network are provided in Appendix A. The code of the proposed method is available on Github[2].

Table 1: Quantitative comparisons (PSNR(dB) / SSIM) of different denoising methods on real datasets. The best zero-shot methods are **bolded** and the second are underlined.

| Category | Method | SIDD Val | SIDD Ben | DND | FMDD |
|---|---|---|---|---|---|
| Non-learning | BM3D [34] | 30.83/.657 | 34.07/.662 | 34.51/.851 | 31.80/.878 |
| Supervised | DnCNN [1] | 37.73/.943 | 37.61/.941 | 37.90/.943 | - |
| Self-supervised | Noise2Void [19] | 27.06/.651 | 26.99/.652 | - | - |
| | NBR2NBR [17] | 27.94/.604 | 27.90/.679 | - | - |
| | CVF-SID [45] | 34.81/.944 | 34.71/.917 | 36.50/.924 | 32.73/.843 |
| | R2R [16] | 35.04/.844 | 35.88/.863 | - | - |
| | APBSN [22] | 36.73/.878 | 36.91/.931 | 38.09/.937 | 31.99/.836 |
| | LG-BPN [23] | 37.32/.886 | 37.28/.936 | 38.43/.942 | - |
| | PUCA [24] | - | 37.54/.936 | 38.83/.942 | - |
| | SS-BSN [25] | - | 37.42/.937 | 38.46/.940 | - |
| | ATBSN [56] | 37.88/.946 | 37.78/.944 | 38.68/.942 | - |
| | SelfFormer [26] | - | 37.69/.937 | 38.92/.943 | - |
| | TBSN [32] | - | 37.78/.940 | 39.08/.945 | - |
| Zero-shot | DIP [27] | 32.11/.740 | - | - | 32.90/.854 |
| | PD-denoising [43] | 33.97/.820 | - | - | 33.01/.856 |
| | Self2Self [28] | 29.46/.595 | 29.51/.651 | - | 30.76/.695 |
| | APBSN-single [22] | 30.90/.818 | 30.71/.869 | - | 28.43/.804 |
| | ZS-N2N [47] | 25.59/.565 | 30.19/.428 | 30.67/.737 | 31.65/.767 |
| | ScoreDVI [50] | 34.75/.856 | 35.39/.859 | 36.19/**.929** | 33.10/.865 |
| | TTT-MIM [51] | 33.22/.638 | 34.91/.805 | **36.91**/- | 31.41/.819 |
| | MASH [29] | 35.06/.851 | 34.80/.814 | 34.75/.910 | 33.71/.882 |
| | Pixel2Pixel [48] | 30.29/- | 31.87/.631 | 34.16/.836 | 33.26/.830 |
| | BSN-INS [30] | 35.31/.868 | 35.05/**.922** | 33.55/.892 | **33.95/.885** |
| | Ours | **36.32/.874** | **36.81**/.874 | 36.79/.924 | 33.47/.882 |

## 4.2 Performance evaluation

**Real dataset:** As a zero-shot method, our method is benchmarked against a comprehensive selection of existing zero-shot techniques. In addition, we also include representative methods from other categories. We use the authors' official code for generating the results or quote published results when available. Table 1 shows that our method achieves the best overall performance among zero-shot approaches, ranking first on two datasets. Notably, the performance gaps to the top performer (0.12 dB and 0.48 dB, respectively) are much smaller than the margins by which our method leads on the other two datasets (1.01 dB and 1.42 dB, respectively). Refer to Appendix F for visual comparisons of the results. Additional validation results on more datasets are provided in Appendix E.2.

**Synthetic dataset:** The noise in the synthetic dataset is not locally correlated, therefore, PD is not applied. This setting can be regarded as an evaluation of the effectiveness of the proposed pixel refilling strategy in addressing the insufficiency of predictive pixels. We only compare our method with the zero-shot methods that perform well on real-world noise. As shown in Table 2, our method achieves competitive performance in terms of PSNR and SSIM. Refer to Appendix G for visual comparisons of the results.

**Runtime and computational overhead by patch matching:** The denoising process for each $256 \times 256$ image from the SIDD dataset takes approximately 60 seconds, with the construction of the cross-scale auxiliary image requiring less than 1 second. This demonstrates that the auxiliary image introduces negligible computational overhead to the overall inference pipeline.

---

[2]https://github.com/QilnnGuo/Denoising-NLR

Table 2: Quantitative comparisons (PSNR(dB) / SSIM) of different zero-shot methods for removing AWGN noise. For natural images, AWGN with $\sigma = 25, 50$ is added. For fastMRI, noise level is $\sigma = 18$ (i.e., $\sigma^2 = 0.005$ in [0,1]). Best results are **bolded** and the second are underlined.

| Method | Kodak24 | | McMaster18 | | fastMRI |
| --- | --- | --- | --- | --- | --- |
| | $\sigma = 25$ | $\sigma = 50$ | $\sigma = 25$ | $\sigma = 50$ | ($\sigma = 18$) |
| DIP [27] | 27.38 / – | 23.95 / – | 27.61 / – | 23.03 / – | 31.32 / – |
| Self2Self [28] | 28.39 / – | 26.22 / – | 28.71 / – | 25.03 / – | 31.99 / – |
| TTT-MIM [51] | **29.67** / – | 23.97 / – | 29.80 / – | 24.01 / – | 29.87 / – |
| Pixel2Pixel [48] | 29.31 / – | 26.26 / – | 29.50 / – | 25.28 / – | 32.25 / **.795** |
| PD-Denoising [43] | 28.98 / .801 | 26.40 / .704 | 28.16 / .784 | 25.96 / .715 | 26.79 / .550 |
| ScoreDVI [50] | 29.31 / .796 | 24.89 / .633 | 29.20 / .788 | 24.78 / .641 | 31.10 / .747 |
| MASH [29] | 24.29 / .593 | 23.63 / .579 | 25.89 / .738 | 25.47 / .703 | 31.35 / .738 |
| BSN-INS [30] | 23.55 / .592 | 22.94 / .551 | 24.27 / .676 | 23.43 / .634 | 29.83 / .713 |
| Ours | 29.12 / **.810** | **26.58** / **.712** | **30.12** / **.853** | **27.41** / **.776** | **32.61** / .783 |

## 4.3 Ablation Study

**Contribution of different components:** There are two techniques involved in the proposed approach: PD [31, 43, 22] and the proposed pixel refilling technique. We evaluate their performance contribution on the SIDD validation dataset with the parameters $\lambda = 0.1$ and $p = 0.5$. As shown in Table 3, when applied individually, the performance improvement from the proposed refilling technique is more modest compared to that from PD, which aligns with our theoretical analysis. However, their improvements arise from different perspectives: PD improves denoising by decorrelating noise but weakening pixel-value correlations, whereas pixel refilling restores pixel-value correlations without introducing additional noise dependence. When combined, they complement each other and achieve the best overall performance.

Table 3: Ablation study of two components.

| PD | Refilling | SIDD (PSNR dB) |
| --- | --- | --- |
| × | × | 31.27 (Plain) |
| × | ✓ | 31.85 (↑ 0.58) |
| ✓ | × | 34.01 (↑ 2.74) |
| ✓ | ✓ | **35.65** (↑ 4.38) |

**Impact of hyperparameter $\lambda$ in loss** (13)**:** In this study, we evaluate the performance impact of the hyperparameter $\lambda$ with the SIDD validation dataset at a mask ratio of 0.5. As reported in Table 4, the optimal result is achieved at $\lambda = 0.2$, whereas other values cause a noticeable performance degradation. Note that pixel-shuffle downsampling is not implemented when applying the proposed method for spatially independent AWGN.

Table 4: Effect of $\lambda$ on SIDD Val.

| $\lambda$ | 0 | 0.1 | 0.2 | 0.5 |
| --- | --- | --- | --- | --- |
| PSNR | 34.95 | 35.65 | **36.11** | 35.72 |

Table 5: Effect of $p$ on SIDD Val.

| $p$ | 0.1 | 0.3 | 0.5 |
| --- | --- | --- | --- |
| PSNR | 36.24 | **36.32** | 36.11 |

**Impact of mask ratio $p$:** The impact of varying the mask ratio $p$, which determines the proportion of masked pixels, is examined on the SIDD validation dataset. As shown in Table 5, performance remains relatively stable across different values of $p$. For AWGN, the ablation study on the effect of the mask ratio is provided in Appendix E.1.

**Ablation on patch matching:** We further study the effectiveness of different scales and patch sizes for patch matching. We fix $\lambda = 0.2, p = 0.3$ and test on SIDD validation dataset for the experiments. For the source of scales, we compare the following settings: same-scale matching, downsampled matching at scales $\downarrow_2$ and $\downarrow_4$. As shown in Table 6, downsampled $\downarrow_2$ achieves the best

result, demonstrating its ability to balance noise reduction and structure preservation. To evaluate the performance impact of different patch sizes, we use scale $\downarrow_2$ and adopt three patch sizes: $4 \times 4$, $8 \times 8$ and $16 \times 16$. As shown in Table 7, patch size $8 \times 8$ achieves the best performance, demonstrating a good trade-off between preserving the pixel correlation and noise decorrelation. More analysis of pixel refilling can be found in Appendix E.

Table 6: Ablation for patches from different scales.

| Choice | Same | $\downarrow_2$ | $\downarrow_4$ |
|--------|------|------|------|
| PSNR | 35.73 | **36.32** | 36.21 |

Table 7: Ablation for different patch sizes.

| Patch Size | 4 | 8 | 16 |
|------------|---|---|----|
| PSNR | 35.88 | **36.32** | 36.14 |

**Inference time comparison:** Table 8 reports the inference time per $256 \times 256$ image, the number of parameters and the FLOPs of several representative zero-shot denoising methods evaluated on the SIDD validation and benchmark datasets. Overall, our method has a smaller model size compared to most methods, while achieving moderate inference time and FLOPs.

Table 8: Inference time, parameter count, and FLOPs comparison.

| Method | Inference Time (s) | Params (M) | FLOPs (G) |
|--------|--------------------|------------|-----------|
| DIP [27] | 89.0 | 13.4 | 31.06 |
| Self2Self [28] | 1273.4 | 1.0 | 9.55 |
| ZS-N2N [47] | 12.5 | 0.02 | 1.45 |
| ScoreDVI [50] | 66.2 | 13.5 | 37.87 |
| Pixel2Pixel [48] | 15.8 | 0.2 | 2.72 |
| MASH [29] | 39.2 | 1.0 | 11.44 |
| BSN-INS [30] | 48.3 | 1.0 | 11.82 |
| Ours | 65.4 | 0.5 | 32.97 |

**Impact of different refilling schemes:** Table 9 examines how different refilling choices affect denoising performance. Our refilling scheme, which both suppresses noise correlation owing to its spatial distance and preserves pixel similarity via cross-scale self-similarity matching, achieves the best results, outperforming both zero-filling (*Vanilla BSD*) and local smoothing (*Mean Filter*).

Table 9: Performance with different pixel refilling schemes

| Choice | Vanilla | Mean Filter | Ours |
|--------|---------|-------------|------|
| PSNR | 34.01 | 35.82 | **36.32** |

### 4.4 Limitation

One key limitation lies in the patch matching process. To serve its purpose of identifying similar patches at a coarser scale, cross-scale similarity needs to hold true, which is the case for natural images, but not so for images in a specific domain. As shown in our experiments, the performance gain on FMDD, a microscopy image dataset, is less significant than that on SIDD, a natural image dataset. In addition, zero-shot requires training sample-specific models, which is computationally expensive when processing a large amount of data.

## 5 Conclusion

In summary, we present a theoretical analysis of the trade-off between noise correlation and pixel correlation in blind-spot denoising, and propose a pixel refilling strategy that effectively balances the two. By leveraging distant but similar pixels with lower noise correlation, our approach addresses both the negative effect of local correlated noise and the issue of insufficient predictive pixels with a large mask ratio. Our method consistently outperforms existing zero-shot BSD methods across multiple benchmarks. In the future, we plan to explore different paradigms for zero-shot denoising, such as leveraging pre-trained models to reduce inference time while maintaining its flexibility.

## Acknowledgments

Hui Ji would like to acknowledge the support from Singapore MOE Academic Research Fund (AcRF) Tier 1 Research Grant A-8000981-00-00.

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

# 6   Appendix

This appendix provides additional material that supports and complements the main paper. Although some of the content may not appear in the main manuscript, it serves to enhance clarity and provide a more complete picture of our method. Specifically, it includes:

## A   Network Architecture Details

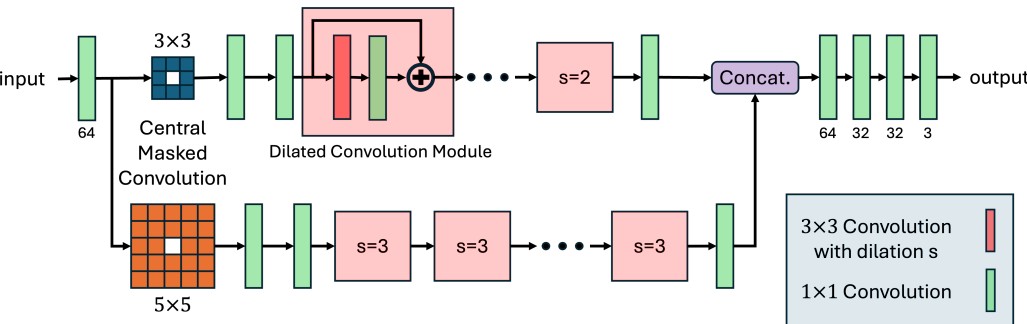

Figure 3: **Visualization of DBSN Architecture.** The architecture is derived from the $3 \times 3$ and $5 \times 5$ central masked convolutional designs introduced in [31], but we adopt a simplified version as proposed in APBSN [22]. Each Dilated Convolution (DC) module consists of a single $3 \times 3$ dilated convolution with a stride $s$, where $s = 2$ is used for the upper path and $s = 3$ for the lower path of the network. We stack 4 DC modules along each path. The number of output channels is indicated below each convolutional layer, with 64 channels used by default. In contrast, the original APBSN [22] configuration uses 9 DC modules and 128 channels by default.

We adopt the simplified D-BSN architecture from APBSN [22], as illustrated in Figure 3. The network we use is shallow and lightweight, comprising only 0.5M parameters, significantly fewer than the 3.7M parameters by default in APBSN [22].

Table 10: Ablation on DBSN model size and computational comparisons on a randomly selected 200-image subset of the SIDD validation.

| PSNR (dB) | [Channels, Blocks] | #Params (M) | FLOPs (G) | Runtime (s) |
|-----------|--------------------|-------------|-----------|-------------|
| 36.56 | [64, 4] | 0.50 | 32.97 | 65.4 |
| 36.55 | [64, 8] | 0.83 | 54.44 | 110.2 |
| 36.58 | [128, 4] | 2.02 | 131.84 | 253.6 |
| 36.65 | [128, 8] | 3.33 | 217.74 | 420.1 |

In addition, Table 10 compares the performance differences among different configurations of DBSN, in terms of model size and computational efficiency. Overall, the performance difference is quite marginal.

## A.1 Ablation on Performance Impact of Network Architecture

Our proposed method is architecture-agnostic and can theoretically enhance any vanilla (zero-filling) BSD framework regardless of the backbone. To verify this, we adopt the UNet architectures used in [14, 29] as backbone models. As shown in Table 11, equipping vanilla UNet-based BSD with our refilling strategy yields a performance gain of more than 2dB, demonstrating the broad applicability of our approach.

Table 11: Ablation on UNet backbone.

|  | SIDD Validation |
| --- | --- |
| Vanilla BSD (UNet backbone) | 33.12 |
| Ours (UNet backbone) | 35.26 |

# B  Proof of Proposition 1 and Corollary 1

**Lemma 1.** *Let* $\mathbf{A} = \alpha\mathbf{I} + \beta\mathbf{E}$*, where* $\alpha > 0$*,* $\beta \geq 0$*, and* $\mathbf{E} = \boldsymbol{e}\boldsymbol{e}^T$ *with* $\boldsymbol{e} = [1,\ldots,1]^T \in \mathbb{R}^M$*. Then the inverse is given by*

$$\mathbf{A}^{-1} = \frac{1}{\alpha}\mathbf{I} - \frac{\beta}{\alpha(\alpha + \beta M)}\mathbf{E},$$

*and* $\mathbf{A}$ *is symmetric positive definite (SPD).*

*Proof of lemma 1.* We verify directly that the proposed inverse satisfies $\mathbf{A}^{-1}\mathbf{A} = \mathbf{I}$. Define

$$\mathbf{A}^{-1} = \frac{1}{\alpha}\mathbf{I} - \frac{\beta}{\alpha(\alpha + \beta M)}\mathbf{E}, \quad \mathbf{A} = \alpha\mathbf{I} + \beta\mathbf{E}.$$

Then,

$$\mathbf{A}^{-1}\mathbf{A} = \left(\frac{1}{\alpha}\mathbf{I} - \frac{\beta}{\alpha(\alpha + \beta M)}\mathbf{E}\right)(\alpha\mathbf{I} + \beta\mathbf{E})$$

$$= \mathbf{I} + \frac{\beta}{\alpha}\mathbf{E} - \frac{\beta}{\alpha + \beta M}\mathbf{E} - \frac{\beta^2 M}{\alpha(\alpha + \beta M)}\mathbf{E}$$

$$= \mathbf{I} + \left(\frac{\beta}{\alpha} - \frac{\beta}{\alpha + \beta M} - \frac{\beta^2 M}{\alpha(\alpha + \beta M)}\right)\mathbf{E}.$$

Now observe:

$$\frac{\beta}{\alpha} - \frac{\beta}{\alpha + \beta M} - \frac{\beta^2 M}{\alpha(\alpha + \beta M)} = \frac{\beta(\alpha + \beta M) - \beta\alpha - \beta^2 M}{\alpha(\alpha + \beta M)} = 0.$$

Therefore, $\mathbf{A}^{-1}\mathbf{A} = \mathbf{I}$, confirming the claimed inverse.

To show $\mathbf{A}$ is symmetric positive definite: symmetry is immediate since both $\mathbf{I}$ and $\mathbf{E}$ are symmetric. For positive definiteness, consider any nonzero $\boldsymbol{x} \in \mathbb{R}^M$:

$$\boldsymbol{x}^T\mathbf{A}\boldsymbol{x} = \alpha\|\boldsymbol{x}\|_2^2 + \beta(\boldsymbol{x}^T\boldsymbol{e})^2.$$

The first term is strictly positive since $\alpha > 0$ and $\boldsymbol{x} \neq \mathbf{0}$; the second term is nonnegative since $\beta \geq 0$. Hence $\boldsymbol{x}^T\mathbf{A}\boldsymbol{x} > 0$ for all $\boldsymbol{x} \neq \mathbf{0}$, so $\mathbf{A}$ is SPD.

**Proposition 1.** *Consider two vectors for weights* $\boldsymbol{a} = [\{a_j\}_{j=1}^M]^T$ *and pixels* $\hat{\boldsymbol{y}} = [\{\hat{y}_j\}_{j=1}^M]^T$*,* $m_j \sim Bernoulli(p)$*. Then with refilled already,* $\hat{y}_j$ *could be rewritten as*

$$\hat{y}_j = m_j y_j + (1 - m_j)\tilde{y}_j. \tag{14}$$

*The linear denoiser is denoted as* $\mathcal{D}_{\boldsymbol{a}}$ *and self-supervised loss is denoted as* $f(\boldsymbol{a})$*,*

$$\mathcal{D}_{\boldsymbol{a}}(\hat{\boldsymbol{y}}) = \boldsymbol{a}^T\hat{\boldsymbol{y}}, \text{ with } \boldsymbol{a}^\star := argmin_{\boldsymbol{a} \in \mathbb{R}^M} f(\boldsymbol{a}), \tag{15}$$

*where*

$$f(\boldsymbol{a}) = \mathbb{E}_{\{m_j\},\{\mu_j\},\{\tilde{\mu}_j\},\{\epsilon_j\},\{\tilde{\epsilon}_j\},n_0,\tilde{n}_0}[\|\mathcal{D}_{\boldsymbol{a}}(\hat{\boldsymbol{y}}) - y_0\|_2^2]. \tag{16}$$

*The prediction risk is defined:*

$$\mathcal{R}_{\boldsymbol{a}} = \mathbb{E}_{\{m_j\},\{\mu_j\},\{\tilde{\mu}_j\},\{\epsilon_j\},\{\tilde{\epsilon}_j\},n_0,\tilde{n}_0}[\|\mathcal{D}_{\boldsymbol{a}}(\hat{\boldsymbol{y}}) - x_0\|_2^2. \tag{17}$$

*Then the prediction risk of optimal denoiser* w.r.t. *self-supervised loss is:*

$$
\begin{aligned}
\mathcal{R}_{\boldsymbol{a}^*} &= -\frac{M(x_0^4 - (p\lambda_n + (1-p)\tilde{\lambda}_n)^2\sigma^4)}{Mx_0^2 + (M-1)\sigma^2(p\lambda_n + (1-p)\tilde{\lambda}_n)^2 + p\sigma_v^2 x_0^2 + (1-p)\tilde{\sigma}_v^2 x_0^2 + \sigma^2} + x_0^2 \\
&= -\frac{Mx_0^4 + \frac{M\sigma^2}{M-1}(Mx_0^2 + p\sigma_v^2 x_0^2 + (1-p)\tilde{\sigma}_v^2 x_0^2 + \sigma^2)}{Mx_0^2 + (M-1)\sigma^2(p\lambda_n + (1-p)\tilde{\lambda}_n)^2 + p\sigma_v^2 x_0^2 + (1-p)\tilde{\sigma}_v^2 x_0^2 + \sigma^2} + \frac{M\sigma^2}{M-1} + x_0^2 \\
&\approx -\frac{x_0^4 - (p\lambda_n + (1-p)\tilde{\lambda}_n)^2\sigma^4}{x_0^2 + \sigma^2(p\lambda_n + (1-p)\tilde{\lambda}_n)^2} + x_0^2 \\
&= x_0^2 + \sigma^2 - \frac{x_0^4 + \sigma^2 x_0^2}{x_0^2 + \sigma^2(p\lambda_n + (1-p)\tilde{\lambda}_n)^2}.
\end{aligned}
\tag{18}
$$

*Proof of proposition 1.* $f(\boldsymbol{a})$ could be rewritten and further simplified (expectation $\mathbb{E}$ will omit the variables $\{m_j\}, \{\mu_j\}, \{\tilde{\mu}_j\}, \{\epsilon_j\}, \{\tilde{\epsilon}_j\}, n_0, \tilde{n}_0$):

$$
\begin{aligned}
f(\boldsymbol{a}) &= \mathbb{E}\left[(\boldsymbol{a}^T\boldsymbol{e}x_0 - \boldsymbol{a}^T\hat{\boldsymbol{\mu}}x_0 + \boldsymbol{a}^T\hat{\boldsymbol{n}} - x_0 - n_0)^2\right] \\
&= \mathbb{E}\left[\boldsymbol{a}^T\boldsymbol{e}\boldsymbol{e}^T\boldsymbol{a}x_0^2 + \boldsymbol{a}^T\hat{\boldsymbol{\mu}}\hat{\boldsymbol{\mu}}^T\boldsymbol{a}x_0^2 + \boldsymbol{a}^T\hat{\boldsymbol{n}}\hat{\boldsymbol{n}}^T\boldsymbol{a} + x_0^2 + n_0^2 - 2\boldsymbol{a}^T\boldsymbol{e}x_0^2 - 2\boldsymbol{a}^T\hat{\boldsymbol{n}}n_0\right] \\
&= \boldsymbol{a}^T\mathbf{E}\boldsymbol{a}x_0^2 + (p\sigma_v^2 + (1-p)\tilde{\sigma}_v^2)\boldsymbol{a}^T\mathbf{I}\boldsymbol{a}x_0^2 \\
&\quad + \boldsymbol{a}^T\left((p\lambda_n + (1-p)\tilde{\lambda}_n)^2\sigma^2)\mathbf{E} + (\sigma^2 - (p\lambda_n + (1-p)\tilde{\lambda}_n)^2\sigma^2)\mathbf{I}\right)\boldsymbol{a} \\
&\quad + x_0 + \sigma^2 - 2\boldsymbol{a}^T\boldsymbol{e}x_0^2 - 2\boldsymbol{a}^T\boldsymbol{e}(p\lambda_n + (1-p)\tilde{\lambda}_n)\sigma^2 \\
&= \boldsymbol{a}^T(x_0^2 + (p\lambda_n + (1-p)\tilde{\lambda}_n)^2\sigma^2)\mathbf{E} \\
&\quad + (p\sigma_v^2 x_0^2 + (1-p)\tilde{\sigma}_v^2 x_0^2 + \sigma^2 - p^2\lambda_n^2\sigma^2 - (1-p)^2\tilde{\lambda}_n^2\sigma^2)\mathbf{I})\boldsymbol{a} \\
&\quad - 2(x_0^2 + (p\lambda_n + (1-p)\tilde{\lambda}_n)\sigma^2)\boldsymbol{a}^T\boldsymbol{e} + x_0^2 + \sigma^2,
\end{aligned}
\tag{19}
$$

where

$$
\begin{aligned}
\hat{\boldsymbol{\mu}}^T &= [\{\hat{\mu}_j\}_j], \quad \hat{\mu}_j = m_j\mu_j + (1-m_j)\tilde{\mu}_j, \\
\hat{\boldsymbol{n}}^T &= [\{\hat{n}\}_j], \quad \hat{n}_j = m_j n_j + (1-m_j)\tilde{n}_j.
\end{aligned}
\tag{20}
$$

Then

$$\mathbb{E}\left[\hat{\boldsymbol{\mu}} \cdot \hat{\boldsymbol{\mu}}^T\right] = (p\sigma_v^2 + (1-p)\tilde{\sigma}_v^2)\mathbf{I}, \tag{21}$$

$$\mathbb{E}\left[\hat{\boldsymbol{n}} \cdot \hat{\boldsymbol{n}}^T\right] = (p\lambda_n + (1-p)\tilde{\lambda}_n)^2\mathbf{E} + (\sigma^2 - (p\lambda_n + (1-p)\tilde{\lambda}_n)^2\sigma^2)\mathbf{I}. \tag{22}$$

Thus

$$f(\boldsymbol{a}) = \boldsymbol{a}^T\mathbf{A}\boldsymbol{a} - 2(x_0^2 + (p\lambda_n + (1-p)\tilde{\lambda}_n)\sigma^2)\boldsymbol{a}^T\boldsymbol{e} + x_0^2, \tag{23}$$

where

$$\mathbf{A} = (x_0^2 + (p\lambda_n + (1-p)\tilde{\lambda}_n)^2\sigma^2)\mathbf{E} + (p\sigma_v^2 x_0^2 + (1-p)\tilde{\sigma}_v^2 x_0^2 + \sigma^2 - (p\lambda_n + (1-p)\tilde{\lambda}_n)^2\sigma^2)\mathbf{I}). \tag{24}$$

We can derive

$$
\begin{aligned}
\boldsymbol{a}^* &= (x_0^2 + (p\lambda_n + (1-p)\tilde{\lambda}_n)\sigma^2)\mathbf{A}^{-1}\boldsymbol{e} \\
&= \frac{x_0^2 + p\lambda_n + (1-p)\tilde{\lambda}_n}{Mx_0^2 + (M-1)\sigma^2(p\lambda_n + (1-p)\tilde{\lambda}_n)^2 + p\sigma_v^2 x_0^2 + (1-p)\tilde{\sigma}_v^2 x_0^2 + \sigma^2}\boldsymbol{e}.
\end{aligned}
\tag{25}
$$

Then we calculate the risk of the optimal linear denoiser and derive

$$
\begin{aligned}
\mathcal{R}_{\boldsymbol{a}^*} &= \mathbb{E}\left[\|\mathcal{D}_{\boldsymbol{a}^*}(\{\hat{y}_j\}_{j=1}^M - x_0\|_2^2\right] = \mathbb{E}\left[(\boldsymbol{a}^{*T}\boldsymbol{e}x_0 - \boldsymbol{a}^{*T}\hat{\boldsymbol{\mu}}x_0 + \boldsymbol{a}^{*T}\hat{\boldsymbol{n}} - x_0)^2\right] \\
&= \boldsymbol{a}^{*T}\mathbf{A}\boldsymbol{a}^* - 2x_0^2\boldsymbol{a}^{*T}\boldsymbol{e} + x_0^2 \\
&= -\frac{M(x_0^4 - (p\lambda_n + (1-p)\tilde{\lambda}_n)^2\sigma^4)}{Mx_0^2 + (M-1)\sigma^2(p\lambda_n + (1-p)\tilde{\lambda}_n)^2 + p\sigma_v^2 x_0^2 + (1-p)\tilde{\sigma}_v^2 x_0^2 + \sigma^2} + x_0^2 \\
&= -\frac{Mx_0^4 + \frac{M\sigma^2}{M-1}(Mx_0^2 + p\sigma_v^2 x_0^2 + (1-p)\tilde{\sigma}_v^2 x_0^2 + \sigma^2)}{Mx_0^2 + (M-1)\sigma^2(p\lambda_n + (1-p)\tilde{\lambda}_n)^2 + p\sigma_v^2 x_0^2 + (1-p)\tilde{\sigma}_v^2 x_0^2 + \sigma^2} + \frac{M\sigma^2}{M-1} + x_0^2 \\
&\approx -\frac{x_0^4 - (p\lambda_n + (1-p)\tilde{\lambda}_n)^2\sigma^4}{x_0^2 + \sigma^2(p\lambda_n + (1-p)\tilde{\lambda}_n)^2} + x_0^2 \\
&= x_0^2 + \sigma^2 - \frac{x_0^4 + \sigma^2 x_0^2}{x_0^2 + \sigma^2(p\lambda_n + (1-p)\tilde{\lambda}_n)^2}.
\end{aligned}
$$

$$(26)$$

**Corollary 1.** *When $\{\tilde{y}_j\}$ satisfies $\tilde{\lambda}_n = \lambda_n$ and $\tilde{\sigma}_v^2 = \sigma_v^2$, the risk of $\mathcal{D}_{\hat{\boldsymbol{a}}^*}$ defined by (15) is:*

$$
\mathcal{R}_{\hat{\boldsymbol{a}}^*} = x_0^2 - \frac{M(x_0^4 - \lambda_n^2\sigma^4)}{Mx_0^2 + (M-1)\sigma^2\lambda_n^2 + \sigma_v^2 x_0^2 + \sigma^2}.
$$

$$(27)$$

*Proof of corollary 1.* Simply replace the $\tilde{\lambda}_n = \lambda_n$ and $\tilde{\sigma}_v = \sigma_v$ in **Proposition 1**, then we could immediately get the result.

## C   Discussion on the Trade-off for the Proposed Linear Model

The effectiveness of the refilling strategy in our linear model depends on the trade-off between noise correlation and pixel-value correlation of the refilled pixels. When the refilled pixels exhibit weaker noise correlation ($\tilde{\lambda}_n$) and reasonably good pixel correlation ($\tilde{\sigma}_v$), the strategy effectively reduces the prediction risk compared to using original pixels alone (the red dash as the baseline from **Corallory 1.**). This scenario is illustrated in Figure 4, where the optimal ratio $p$ lies within $(0, 1)$, demonstrating a meaningful balance between original and refilled pixels.

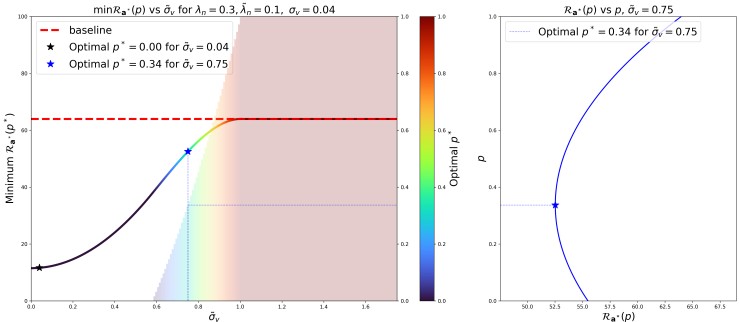

Figure 4: With reasonable parameter settings: $M = 127, x_0 = 100, \lambda_n = 0.3, \sigma_v = 0.04, \tilde{\lambda}_n = 0.1$, the left subfigure illustrates one example of our linear model. In this plot, the rainbow-colored histogram corresponds to the optimal resampling ratio $p^*$ for each value of $\tilde{\sigma}_v$. The right subfigure shows a special case, highlighting how the ratio $p$ influences the prediction risk when the pixel correlation is reduced, i.e., $\sigma_v < \tilde{\sigma}_v = 0.75$.

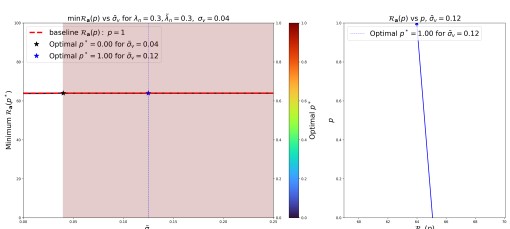

(a) No noise decorrelation and pixel correlation too weak, right subfigure suggests no refilling;

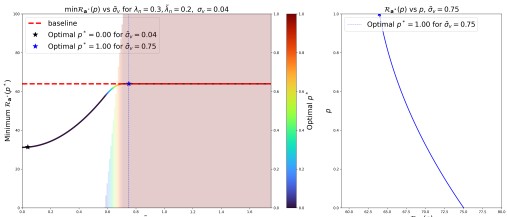

(b) Good noise decorrelation but pixel correlation too weak, right subfigure suggests no refilling;

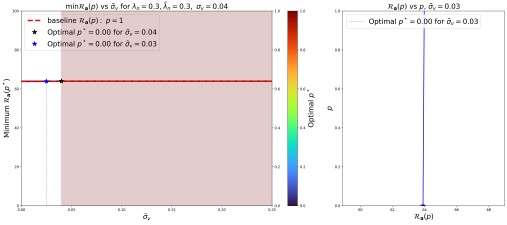

(c) No noise decorrelation and stronger pixel correlation, right subfigure suggests refilling all pixels;

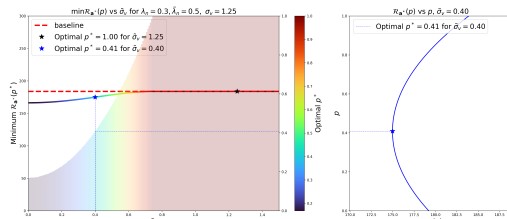

(d) Slightly larger noise correlation dominates, requiring strong pixel correlation to make refilling strategy work.

Figure 5: Prediction risk curves under different bad noise or signal correlation settings.

However, when the refilled pixels fail to suppress noise correlation or badly suffer from poor signal correlation, the strategy becomes ineffective, and the optimal solution degenerates to $p \approx 1$, meaning the model relies entirely on original pixels. The failure modes are shown in Figure 5 (a) (b) and (c). As we have stated that the noise correlation dominates when $M$ is quiet large, if the noise correlation is slightly larger in the refilled pixels, only when the refilling pixels exhibit terribly strong pixel

correlation could the refilling strategy work, see Figure 5 (d). These findings support our theoretical analysis and emphasize the importance of matching refilled pixels with both low noise correlation and adequate signal fidelity.

## D  Visualization of Patch Matching

See how patch matching works on different scales in Figure 6. Note that when patch matching within the same scale, the patch $P_i$ is replaced by $\widetilde{P}_j$ with the smallest $\ell_2$-distance $\|P_i - \widetilde{P}_j\|_2$ among all patches except the one with the same center as $P_i$ in $\widetilde{Y}$, i.e., second smallest $\ell_2$-distance.

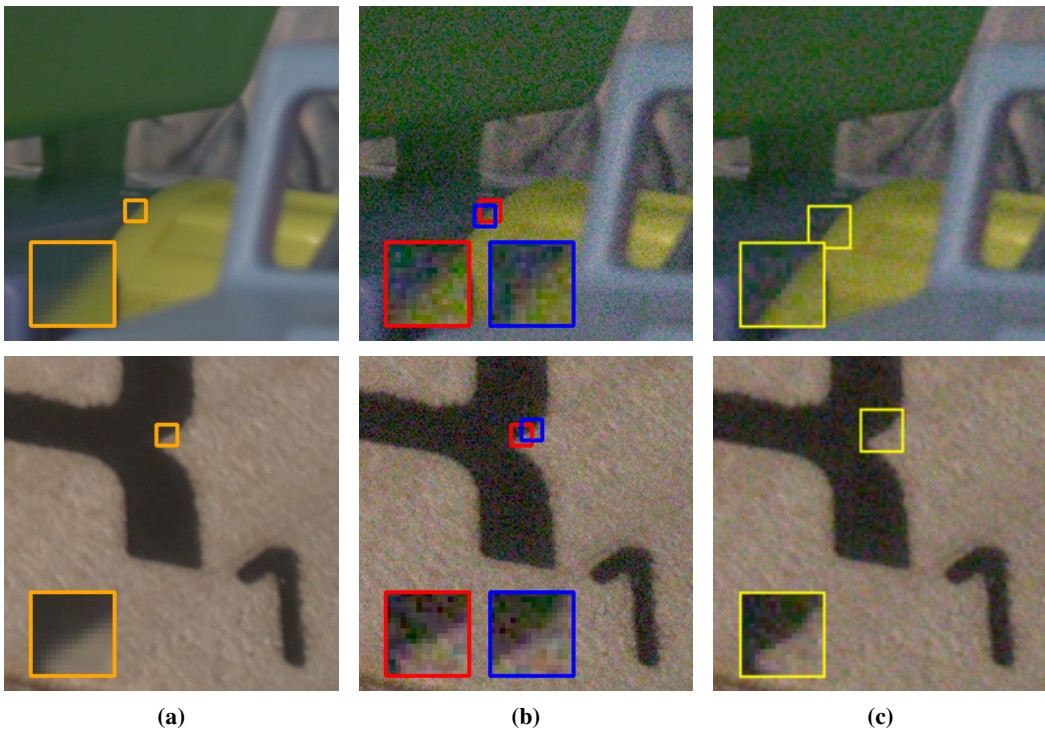

(a)         (b)         (c)

Figure 6: (a) Clean image with a reference patch highlighted in orange. (b) The corresponding noisy image patch (red) at the same spatial location, together with its most similar patch (blue) found within the same noisy image. (c) The noisy image at a coarser scale (obtained by $2 \times 2$ average pooling), where the most similar patch (yellow) corresponds to the noisy reference patch (red).

## E  More Analysis on Pixel Refilling

In this section, we first ablate the mask ratio for the AWGN case, followed by additional experiments to further verify the robustness of our strategy. Specifically, we evaluate performance on additional real-world datasets, and examine the robustness under various noise levels against vanilla BSD (zero-filling). All experiments use the lightweight DBSN model with a channel configuration of [64, 4] as the backbone.

Table 12: Ablation of mask ratio on McMaster18 with AWGN $\sigma = 50$.

| $p$ | 0.3 | 0.5 | 0.7 | 0.9 |
|---|---|---|---|---|
| PSNR | 27.37 | **27.39** | 27.37 | 27.19 |

### E.1 Ablation on Mask Ratio for AWGN

We examine the performance impact of varying the mask ratio $p$, which controls the proportion of masked pixels, under AWGN with $\sigma = 50$. As shown in Table 12, performance remains stable across different values of $p$, and we adopt $p = 0.5$.

### E.2 Evaluation on More Real Datasets

PolyU [57] contains 40 pairs of noisy-clean images captured at various ISO levels. These datasets span a wide range of noise levels. NIND [58] is another real-world dataset consisting of clean-noisy image pairs captured at ISO levels of 3200, 4000, 5000, and 6400, with 22, 14, 13, and 79 pairs, respectively. We selected two sub-datasets to represent two types of noise level, ISO3200 and ISO5000, for comparison. It can be seen that, on the PolyU dataset which features relatively low noise level, our method still achieves the best performance. This suggests that the proposed pixel refilling strategy remains effective even when the noise level is low. Note that the results on PolyU are quoted from Pixel2Pixel [48].

Table 13: Comparison of zero-shot denoising methods on PolyU and NIND datasets. Results are reported as PSNR(dB) / SSIM.

| Method | PolyU | NIND ISO3200 | NIND ISO5000 |
|---|---|---|---|
| ZS-N2N | 35.17/– | 29.40/0.642 | 26.58/0.546 |
| Pixel2Pixel | 36.11/– | 32.30/0.783 | 30.18/0.745 |
| ScoreDVI | 34.00/– | 34.12/0.845 | 31.64/0.803 |
| MASH | 31.97/– | 33.32/0.834 | 31.34/0.802 |
| Ours | **36.29/0.942** | **34.40/0.860** | **32.58/0.835** |

### E.3 Robustness w.r.t. Different Noise Levels

To evaluate the robustness of the proposed pixel refilling strategy with respect to different noise levels, we conduct experiments under different levels of AWGN, including low ($\sigma = 10$), moderate ($\sigma = 25$), and extreme ($\sigma = 100$), using the same backbone and mask ratio as vanilla BSD. The results show that under extreme noise conditions, the effectiveness of pixel refilling becomes weaker, as severe corruption makes reliable texture matching more difficult. Nevertheless, in all cases, the pixel refilling strategy consistently outperforms its vanilla counterpart.

Table 14: Comparison on Kodak24 under different noise levels ($\sigma = 10, 25, 100$).

| Method | $\sigma = 10$ (Low) | | $\sigma = 25$ (Moderate) | | $\sigma = 100$ (Extreme) |
|---|---|---|---|---|---|
| Ratio of masked pixels | 0.5 | 0.1 | 0.5 | 0.1 | 0.1 |
| Vanilla BSD | 29.75 | 30.96 | 27.91 | 28.77 | 23.26 |
| Ours | **30.98** | **31.39** | **29.03** | **29.12** | **23.36** |

# F Visual Comparisons for Real-World Denoising Experiments

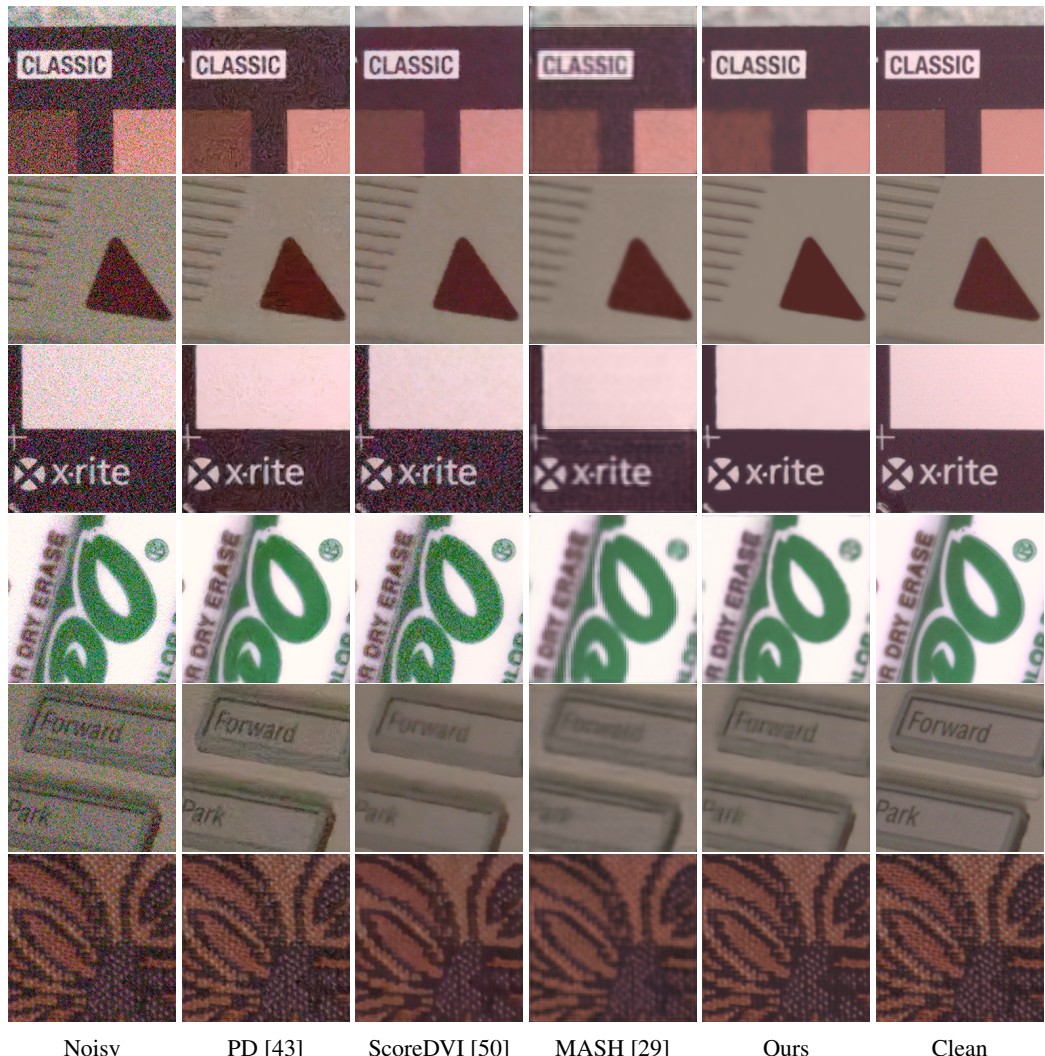

| Noisy | PD [43] | ScoreDVI [50] | MASH [29] | Ours | Clean |

Figure 7: Qualitative comparisons on **SIDD Validation** images. Each row corresponds to a different noisy input, and columns show outputs from different denoising methods: noisy input, PD-denoising [43], ScoreDVI [50], MASH [29], our method, and the clean reference. Our method consistently preserves more fine details and recovers textures more faithfully compared to other approaches.

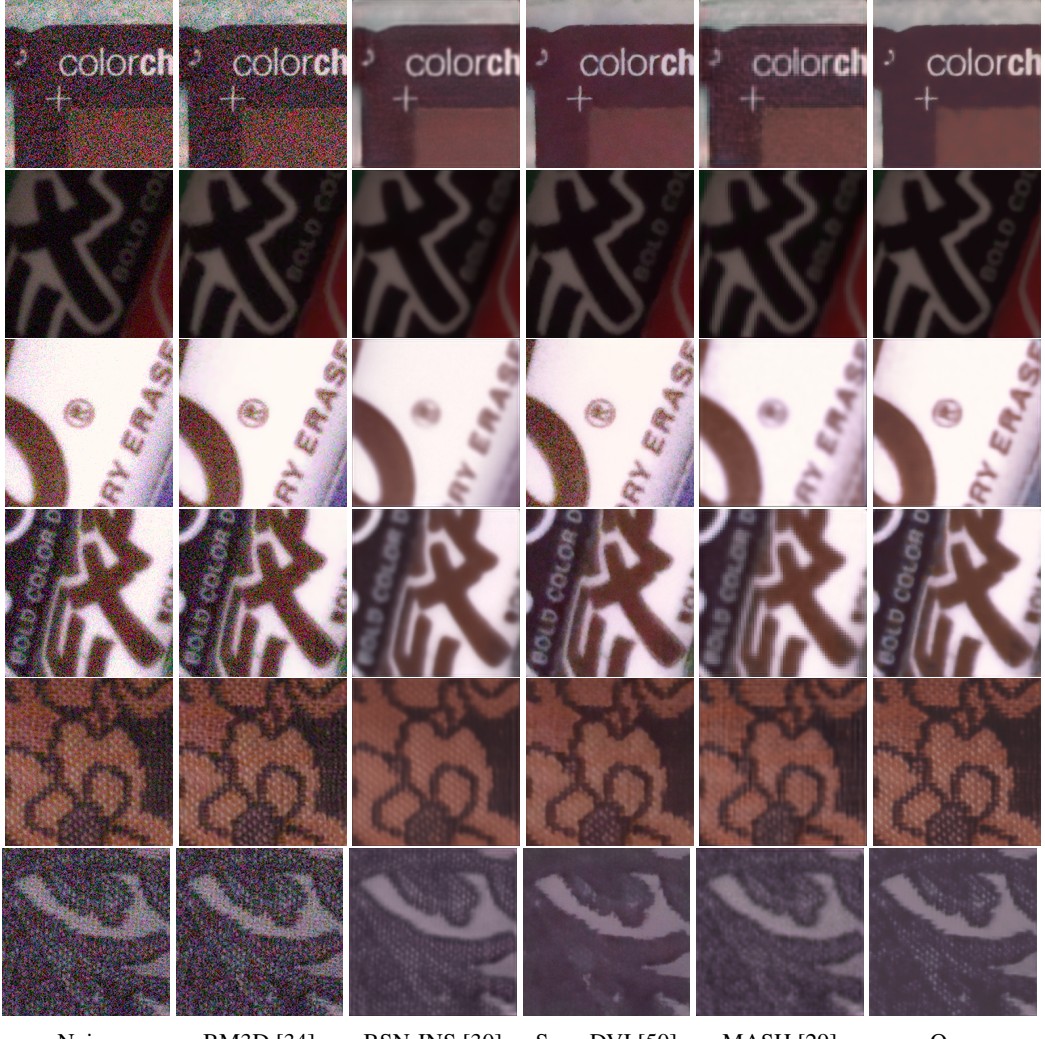

Noisy      BM3D [34]      BSN-INS [30]      ScoreDVI [50]      MASH [29]      Ours

Figure 8: Qualitative comparisons on **SIDD Benchmark** images. Each row corresponds to a different noisy input, and columns show outputs from different denoising methods: noisy input, BM3D [34], BSN-INS [30], ScoreDVI [50], MASH [29], our method. Our method maintains visual fidelity and reduces noise more effectively than other zero-shot methods.

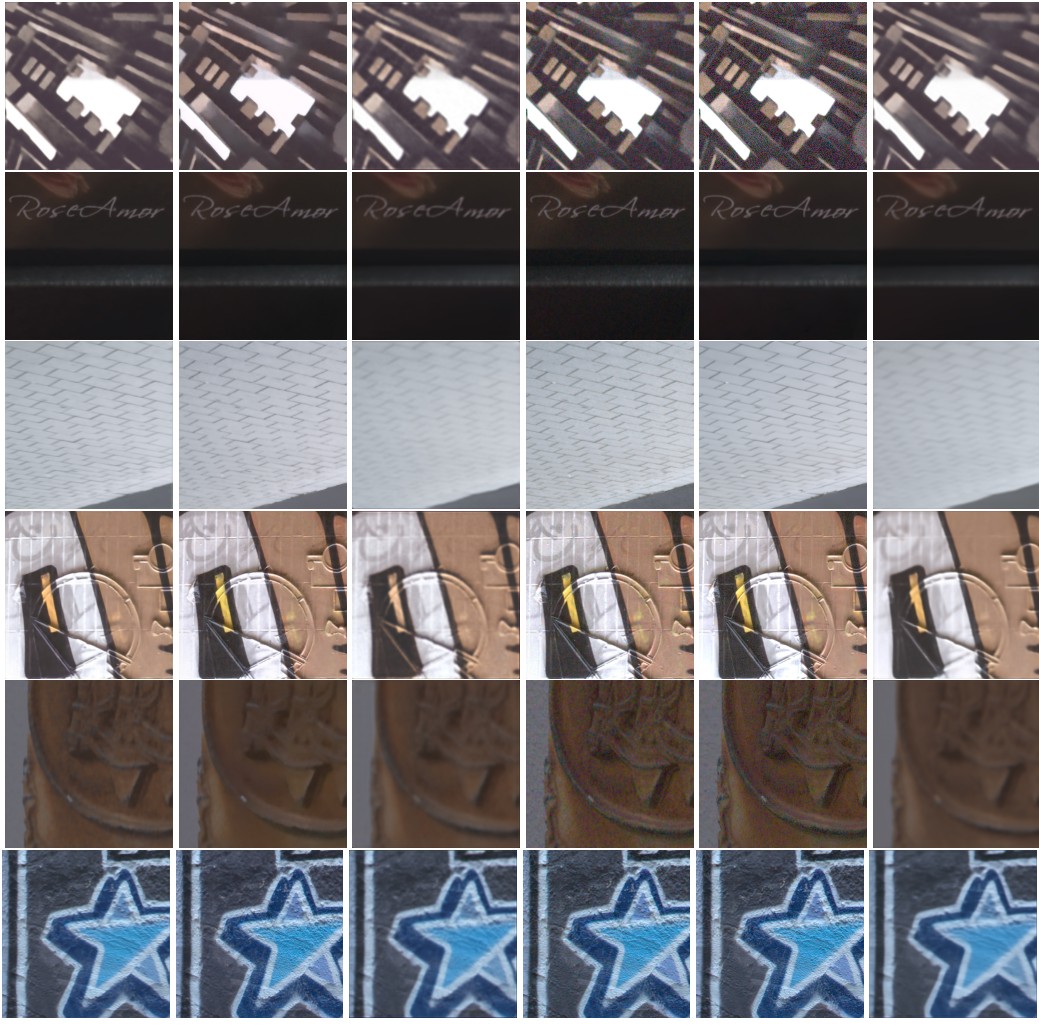

Ours      ScoreDVI [50]      MASH [29]      Pixel2Pixel [48]      TTT-MIM [51]      BSN-INS [30]

Figure 9: Qualitative comparisons on **DND Benchmark** images. Each row corresponds to a different noisy input, and columns show outputs from different denoising methods: Ours, ScoreDVI [50], MASH [29], Pixel2Pixel [48], TTT-MIM [51] and BSN-INS [30]. Our method maintains visual fidelity and reduces noise more effectively than other zero-shot methods.

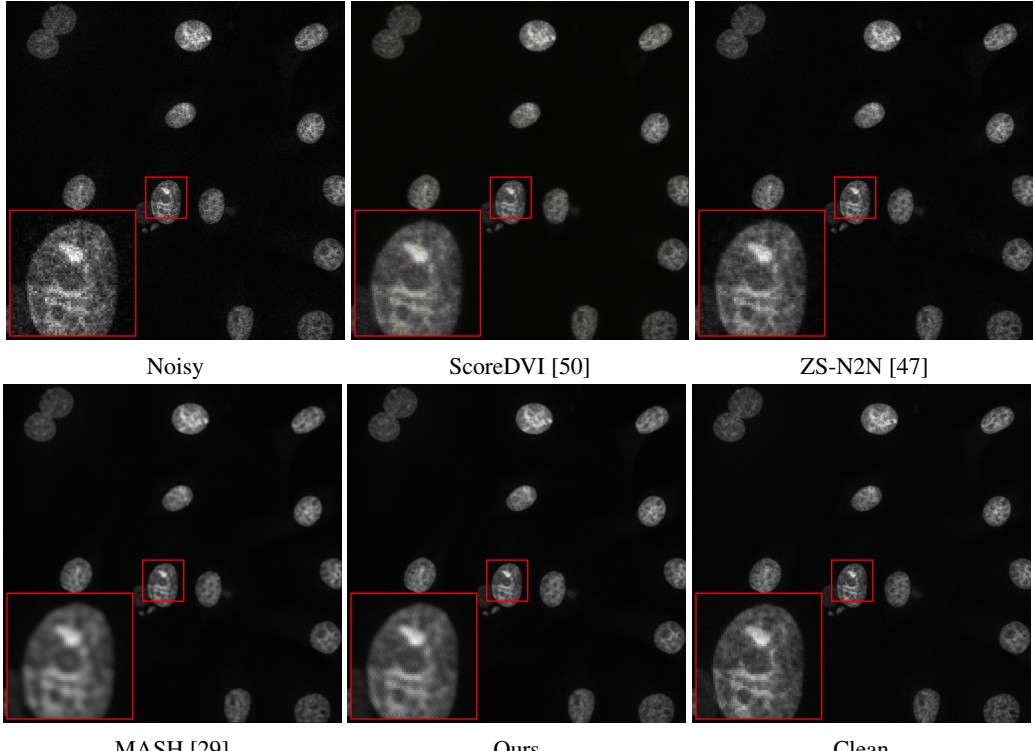

Figure 10: Qualitative comparison on the **FMDD**.

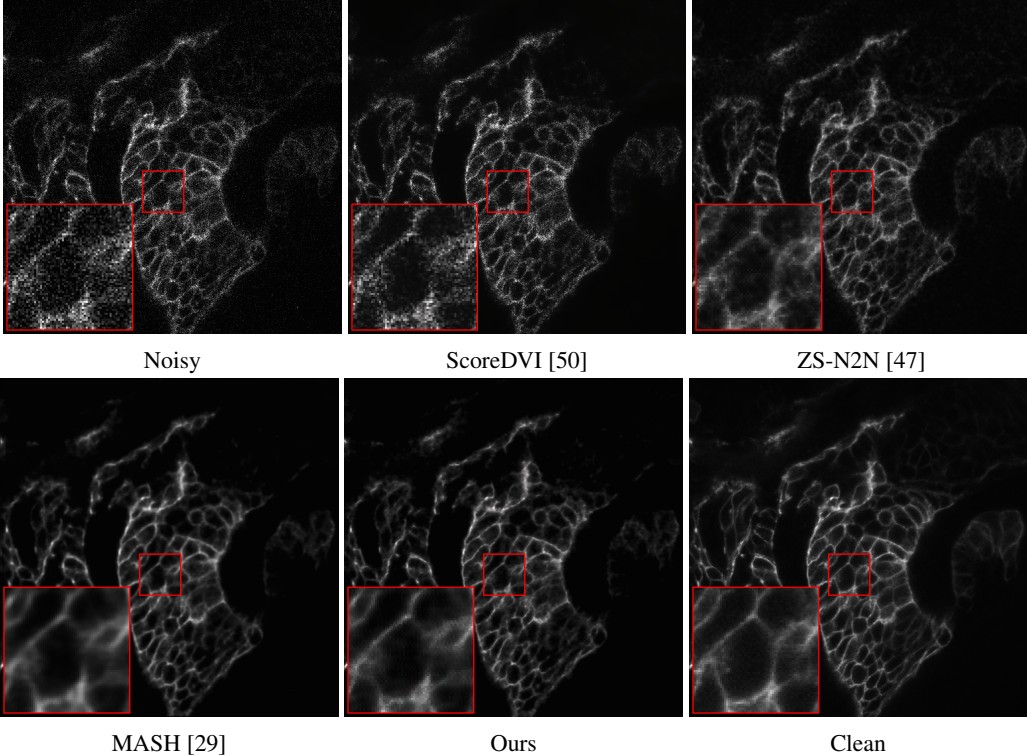

Figure 11: Qualitative comparison on the **FMDD**.

# G Visual Comparisons for Synthetic Denoising Experiments

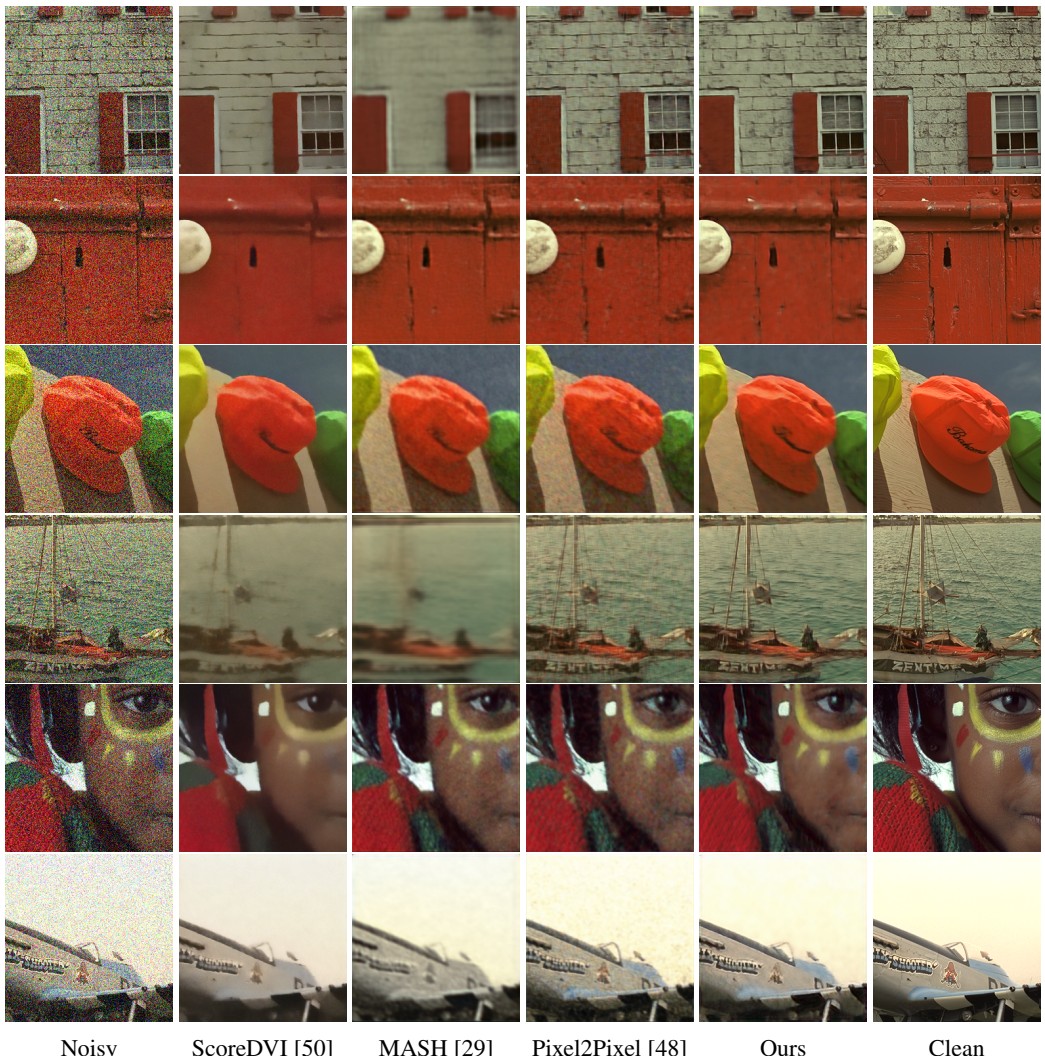

Noisy  ScoreDVI [50]  MASH [29]  Pixel2Pixel [48]  Ours  Clean

Figure 12: Qualitative comparisons on **Kodak** with AWGN $\sigma = 50$. Each row corresponds to a different noisy input, and columns show outputs from different denoising methods: Noisy input, ScoreDVI [50], MASH [29], Pixel2Pixel [48], our method, and the clean reference. Our method consistently preserves more fine details and recovers textures more faithfully compared to other approaches.

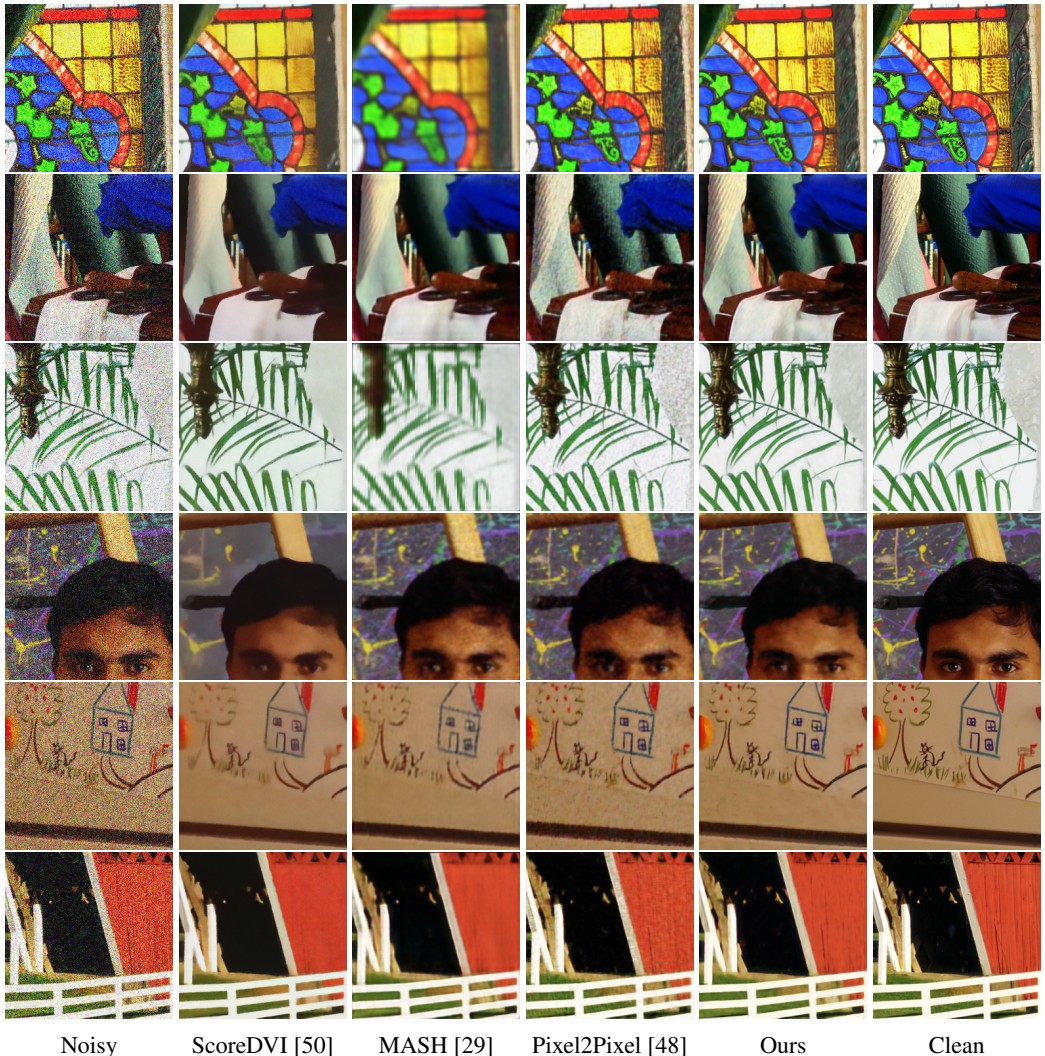

| Noisy | ScoreDVI [50] | MASH [29] | Pixel2Pixel [48] | Ours | Clean |

Figure 13: Qualitative comparisons on McM with AWGN $\sigma = 50$. Each row corresponds to a different noisy input, and columns show outputs from different denoising methods: Noisy input, ScoreDVI [50], MASH [29], Pixel2Pixel [48], our method, and the clean reference. Our method consistently preserves more fine details and recovers textures more faithfully compared to other approaches.

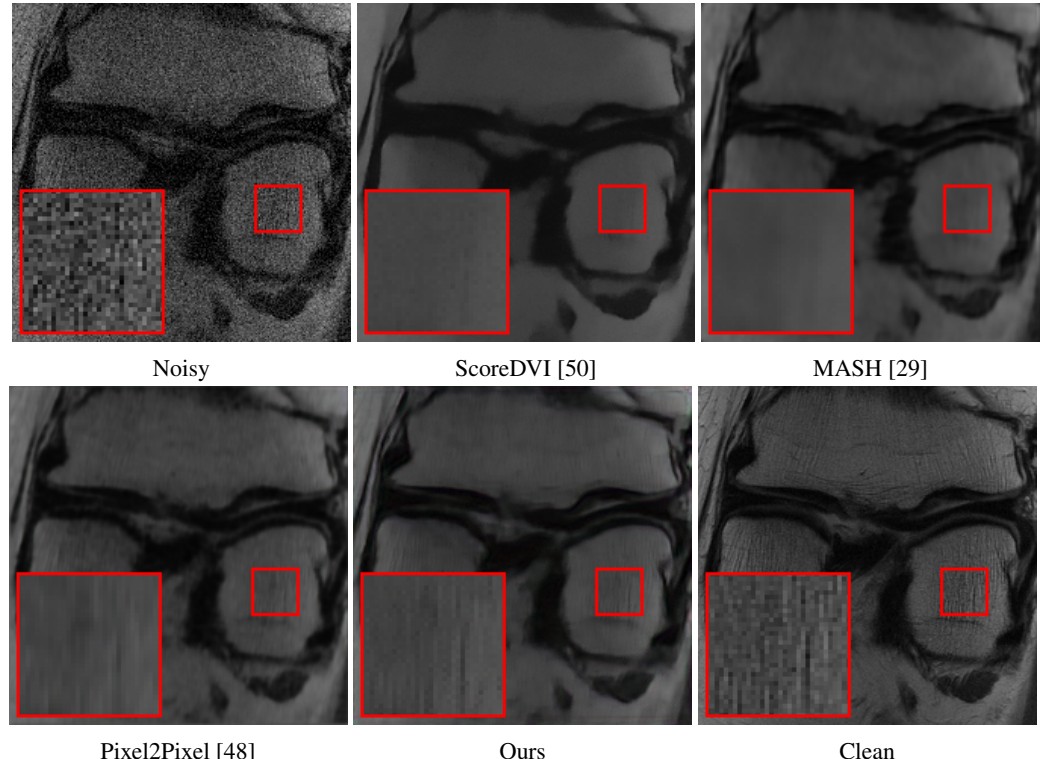

Figure 14: Qualitative comparison on one slice from the **fastMRI** Dataset with AWGN $\sigma = 18$. Our result shows more consistent textures than other zero-shot denoising methods.

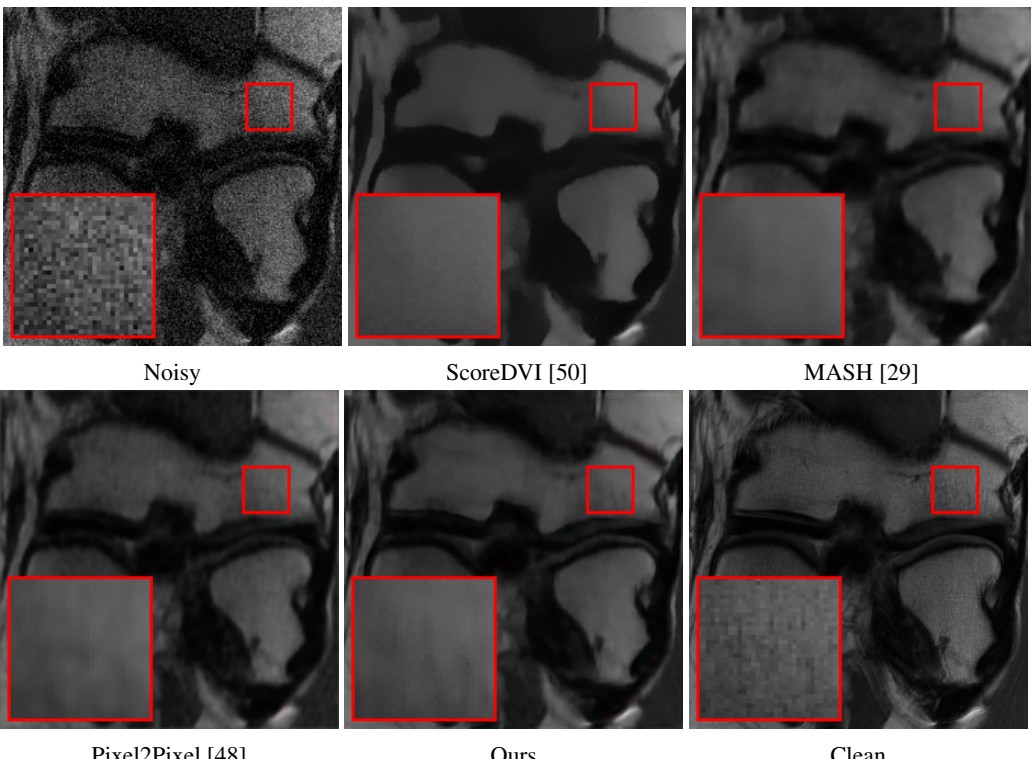

Figure 15: Qualitative comparison on one slice from the **fastMRI** Dataset with AWGN $\sigma = 18$. Our result shows more consistent textures than other zero-shot denoising methods.

