# OpenReview forum: "Zero-Shot Blind-Spot Image Denoising via Cross-Scale Non-Local Pixel Refilling"
_NeurIPS.cc/2025/Conference — NeurIPS 2025 poster_

### Official Review · Reviewer_A71X · 2025-06-26

**Clarity:** 3
**Significance:** 3
**Originality:** 2
**Rating:** 4
**Confidence:** 5

**Summary:**

The paper highlights the limitations of blind-spot networks (BSNs) for zero-shot image denoising in the presence of locally correlated noise, which is the case with real-world camera or microscope noise. It introduces a theoretical trade-off between noise correlation and structural similarity when replacing masked pixels. Based on this, the authors propose a method that substitutes masked pixels with non-local, but visually similar ones to reduce noise correlation while preserving structure. Experiments show improved denoising performance compared to existing methods.

**Questions:**

N/A

**Ethical Concerns:**

["NO or VERY MINOR ethics concerns only"]

**Final Justification:**

The rebuttal clarified a lot of my concerns, specially the similarity to Pixel2Pixel, and the complexity of the proposed method. Overall, based on the extra experiments provided in the rebuttal, the method seems to achieve good performance compared to existing work while having moderate computational complexity. After reading other reviews, I recommend acceptance considering the authors will add the extra experiments from the rebuttal in the final paper.

**Limitations:**

yes

**Quality:**

2

**Strengths And Weaknesses:**

Strengths:
- Clear motivation, good results, several recent baselines used for comparisons

Weaknesses:
- The proposed method seems similar to Pixel2Pixel, which also replaces masked pixels with the most similar pixels from non-local regions. A dedicated section in the paper with clear explanation of how the proposed method differs from Pixel2Pixel would help make the contribution and novelty more appreciated.
- As Pixel2Pixel is also made for solving the problem of correlated noise, it should be used as a baseline in Table 1 for the real-world noise, having it as a baseline in Table 2 is not sufficient.
- More on missing comparisons: Since ZS-N2N works well for pixel wise independent noise, it should be used as a baseline in Table 2, as the noise added there is AWGN.
- A concern for zero shot methods is the denoising time, since a network is trained for each image. Therefore a comparison of the denoising speed/compute with competing zero-shot methods is essential.
- Table 1 would benefit from having the DND as a benchmark dataset, since the paper's main contribution is denoising correlated noise, which is prevalent in real-world camera noise, and DND is a standard benchmark for that.
- It is common to have an intermediate noise level for AWGN denoising with $\sigma=$ 25 or 30, which is missing in Table 2.
- Missing description of the architecture/model size of the NN used in the paper.
- Many typos and grammatical mistakes such as "(BSD) method offer" line 1, "an noisy" line 240, "we also uses" line 252, "Our method consistently outperforming existing" line 322, and several others.
- No code provided

---

> ### Author Rebuttal · Authors · 2025-07-30
>
> Thank you for the constructive feedback. Our method builds on the Blind-Spot Denoising (BSD) framework, a fundamental approach in recent truth-free denoising methods, and introduces a novel pixel refilling strategy. We also provide, to the best of our knowledge, the first theoretical analysis of BSD, offering insights into its assumptions and implications for practical implementation. See below for our clarifications and requested additional experiments. All these clarifications and experimental results will be included in the revision.
>
> ---
>
> **[W1]** *Key Differences from Pixel2Pixel and Our Contributions*
>
> Thank you to the reviewer for pointing out the relevance of Pixel2Pixel. We will include it in the revised manuscript and provide a detailed discussion.
>
>
>
> While both methods aim to address noise correlation in self-supervised denoising, they are grounded in two different frameworks. Pixel2Pixel is built upon the Noise2Noise (N2N) paradigm [1],
>
> $$
> \min_\phi \left\\| \mathcal{D}_\phi(\mathbf{y}_1) - \mathbf{y}_2 \right\\|_2^2  \quad (\text{Noise2Noise}),
> $$
>
> where its aim is to generate image pairs with uncorrelated noise that approximate noise-clean pairs for defining the loss function.
>
> In contrast, our method builds on the Blind-Spot Denoising (BSD) framework:
>
> $$
> \min_\phi\\|(1-M)\odot[\mathcal{D}_\phi(M\odot \mathbf{y})-\mathbf{y}]\\|_2^2 \quad (\text{Blind-spot}).
> $$
>
>  where $M$ is the zero mask and the NN $\mathcal{D}_\phi(\cdot)$ is trained to predict masked-out pixels from un-masked-out pixels.
>
>
> While both approaches leverage patch similarity to mitigate noise correlation between input-output pairs, Pixel2Pixel uses matched patches to construct pseudo image pairs for Noise2Noise-style training, and is more closely related to methods like ZS-N2N[2], R2R[3], and NBR2NBR[4]. In contrast, our method employs matched patches to refill masked pixels within the BSD  framework. Importantly, our approach is supported by a mathematical analysis that provides theoretical justification for the design, offering a deeper understanding of BSD and paving the way for future advancements in this framework.
>
>
> [1] *Lehtinen et al. (2018). Noise2Noise: Learning image restoration without clean data. (ICML).*
>
> [2] *Mansour et al. (2023). Zero-shot Noise2Noise: Efficient image denoising without any data. (CVPR).*
>
> [3] *Pang et al. (2021). Recorrupted-to-Recorrupted: Unsupervised deep learning for image denoising. (CVPR).*
>
> [4] *Huang et al. (2021). Neighbor2Neighbor: Self-supervised denoising from single noisy images. (CVPR).*
>
> ---
>
> **[W2]** *Comparison to pixel2pixel in Table 1*
>
> See the table below for the comparison to pixel2pixel, where the results of pixel2pixels on SIDD validation and PolyU are cited from the paper and others are generated using the authors' code. Our method outperformed pixel2pixel by a large margin on all datasets.
>
> *Table D.2.1. Comparison of zero-shot denoising methods on different real denoising datasets.*
>
> |Method|DND|PolyU|NIND ISO3200|NIND ISO5000|SIDD Val|SIDD Ben|
> |-|-|-|-|-|-|-|
> |ZS-N2N| 30.67/0.737| 35.17/--|29.40/0.642|26.58/0.546|25.59/0.565|30.19/0.428|
> |Pixel2Pixel|34.16/0.836|*36.11*/--|32.30/0.783|30.18/0.745|30.29/--|31.87/0.631|
> |ScoreDVI|*36.19/0.929*|34.00/--|34.12/0.845|31.64/0.803|34.75/0.857|*35.39/0.859*|
> |MASH|34.75/0.910|31.97/--|*33.32/0.834*|*31.34/0.802*|*35.06/0.851*|34.80/0.814|
> |Ours|**36.79/0.924**|**36.29/0.942**|**34.40/0.860**|**32.58/0.835**|**36.32/0.874**|**36.81/0.874**|
>
>
> where we also include DND[5], PolyU[6] and NIND[7] subsets results for comprehensiveness. DND[5] consists of 50 noisy-clean pairs, formed by shooting the same scene twice with different ISO values. PolyU[6] contains 40 pairs of noisy-clean images captured at various ISO levels, and NIND[7] is a real-world dataset consisting of clean-noisy image pairs captured at ISO levels of 3200, 4000, 5000, and 6400, with 22, 14, 13, and 79 pairs, respectively, where we selected ISO 3200 and ISO 5000 subsets for comparison.
>
>
> [5] *Plotz et al. (2017). Benchmarking denoising algorithms with real photographs. (CVPR).*
>
> [6] *Xu et al. (2018). Real-world noisy image denoising: A new benchmark. (arXiv).*
>
> [7] *Brummer et al. (2019). Natural image noise dataset. (CVPRW).*
>
> ---
>
> **[W3]** *Comparison to ZS-N2N in Table 2*
>
> See the table below for the comparison. The results of ZS-N2N are directly quoted from the paper of pixel2pixel. Our method outperformed ZS-N2N by a large margin, as shown in the table below.
>
>
> *Table D.3.1. Comparison of PSNR results (dB) under Gaussian noise*
>
> | Method       | fastMRI σ=18 | McMaster18 σ=25 | McMaster18 σ=50 | Kodak24 σ=25 | Kodak24 σ=50 |
> |--------------|---------------|------------------|------------------|----------------|----------------|
> | DIP          | 31.32         | 27.61           | 23.03           | 27.38         | 23.95         |
> | S2S          | 31.99         | 28.71           | 25.03           | 28.39         | 26.22         |
> | ZS-N2N       | 32.19         | 28.80           | 24.02           | 29.07         | 24.81         |
> | Pixel2Pixel  | 32.25       | 29.50         | 25.28         | *29.31*    | 26.26       |
> | **Ours**     | *32.61*    | *30.12*       | *27.41*       | 29.12       | *26.58*     |
>
> ---
>
>
> **[W4]**  *Comparison in terms of speed*
>
> Please see the table below for the detailed comparison of inference time, parameter count, and FLOPs of competing zero-shot methods.
>
> *Table D.4.1. Complexity comparison of zero-shot methods.*
>
> | Method                          | Infer. time (s) | Params (M) | FLOPs (G) |
> |--------------------------------|------------------|------------|-----------|
> | DIP                            | 146.2            | 13.4       | 31.06     |
> | PD-denoising                   | 0.1              | 0.7        | 46.94     |
> | ZS-N2N                         | 9.8              | 0.02        | 1.45    |
> | Pixel2Pixel                    | 24.3               | 0.2        | 2.72     |
> | Self2Self                      | 3546.5           | 1.0        | 9.55      |
> | ScoreDVI                       | 81.2             | 13.5       | 37.87     |
> | MASH                           | 75.3             | 1.0        | 11.44     |
> | **Ours**                       | 65.4        | 0.5    | 32.97 |
>
>
> We have detailed a comparison of inference time, parameter count, and FLOPs in Table D.4.1. Our method achieves a favorable balance between performance and computational efficiency among zero-shot denoising methods. While ZS-N2N and Pixel2Pixel are faster, they suffer from a significant drop in performance. On the other hand, ScoreDVI and MASH have performance closer to ours but with comparable computational complexity.
>
> ---
>
> **[W5]** *Table 1 would benefit from having the DND as a benchmark dataset*
>
> To date, no existing zero-shot methods have reported performance on DND. Due to time constraints, we only compared competing zero-shot methods exclusively on DND. As shown in the table below, our method remains the top performer among them.
>
>
> *Table D.5.1. Comparison of zero-shot denoising methods on different real denoising datasets.*
>
> |Method|DND|PolyU|NIND ISO3200|NIND ISO5000|SIDD Val|SIDD Ben|
> |-|-|-|-|-|-|-|
> |ZS-N2N| 30.67/0.74| 35.17/--|29.40/0.642|26.58/0.546|25.59/0.565|30.19/0.428|
> |Pixel2Pixel|34.16/0.84|*36.11*/--|32.30/0.783|30.18/0.745|30.29/--|31.87/0.631|
> |ScoreDVI|*36.19/0.929*|34.00/--|34.12/0.845|31.64/0.803|34.75/0.857|*35.39/0.859*|
> |MASH|34.75/0.910|31.97/--|*33.32/0.834*|*31.34/0.802*|*35.06/0.851*|34.80/0.814|
> |Ours|**36.79/0.924**|**36.29/0.942**|**34.40/0.860**|**32.58/0.835**|**36.32/0.874**|**36.81/0.874**|
>
>
> ---
>
> **[W6]** *intermediate noise level for AWGN denoising with 25 or 30 in Table 2*
>
> Please refer to Table D.3.1 in our response to Q3.
>
> ---
> **[W7]** *Missing description of the architecture/model size of the NN used in the paper*
>
> Please refer to Appendix A of the supplemental material for the details of the architecture. The NN we use is a small DBSN with 64 channels and 4 blocks, which only has 0.5M parameters.
>
> ---
> **[W8]** *Many typos and grammatical mistakes*
>
> Thank you for pointing out the grammatical errors and typos. We will thoroughly proofread the manuscript and correct all identified typos and grammatical mistakes.
>
> ---
> **[W9]** *No code provided*
>
> We will release the full code upon acceptance of the paper.
>
> ---
>
> Please let us know if you have any further questions. Thank you again for the helpful feedback.

---

> ### Comment · Reviewer_A71X · 2025-08-05
>
> I thank the authors for their detailed response, which has addressed most of my concerns, and I am therefore inclined to increase my score.
>
> Particularly my main concern, which is the similarity to Pixel2Pixel has been resolved with the extra experiments in Table D.2.1, as it shows that the proposed method significantly outperforms Pixel2Pixel for natural noise.
>
> Moreover, the results in Table D.4.1, show that the computational complexity of the proposed method is not too high, and that it comes at an acceptable cost given the good performance.
>
> For the sake of completion, the recent baseline TTT-MIM that the authors already have in their paper should be added to Tables D.2.1 and D.4.1. It should be relatively straightforward to add its complexity measurements to D.4.1, and regarding D.2.1, the authors already have some results in the paper, and others such as DND can be read directly from the TTT-MIM paper (since DND is evaluated through the official website, there is no mismatch in evaluation).
>
> Overall, the rebuttal has clarified a lot, and I highly encourage the authors to include these extra results from the rebuttal to the main paper (specially D.2.1 and D.4.1), as these make the paper's contribution more significant.

---

> > ### Author Response · Authors · 2025-08-06
> > **Response with thanks**
> >
> > We sincerely thank the reviewer for the thoughtful follow-up and the positive reassessment of our submission.
> >
> > We are pleased that the additional experiments have addressed the main concern regarding the similarity to Pixel2Pixel and helped clarify the computational efficiency of our method. We also greatly appreciate the suggestion to include TTT-MIM in Tables D.2.1 and D.4.1 for completeness. We agree that this addition would further strengthen the paper and confirm that we will incorporate the relevant complexity metrics and performance results, along with other additional experiments and clarifications, into the revised manuscript.
> >
> > Once again, we thank the reviewer for the constructive feedback and helpful suggestions, which we will carefully reflect in the revision.

---

### Official Review · Reviewer_N84f · 2025-07-02

**Clarity:** 3
**Significance:** 2
**Originality:** 2
**Rating:** 4
**Confidence:** 5

**Summary:**

This paper introduces a zero-shot blind-spot denoising (BSD) method via pixel-refilling. Existing mask-based real-world denoising methods could break noise correlation, but simultaneously damage the content. Instead, the authors theoretically analysis and address two major challenges in BSD: limited predictive pixels due to masking and biased loss caused by local noise correlation. Therefore, they propose a novel pixel refilling strategy for BSD method that filling masked pixels with carefully selected noisy pixels from the same image with lower noise correlation and sufficient pixel correlation. Experiments in real and synthetic noisy images demonstrate the effectiveness of proposed approach.

**Questions:**

Please see Weaknesses. If the questions are addressed well, I will increase the rating.

**Ethical Concerns:**

["NO or VERY MINOR ethics concerns only"]

**Final Justification:**

My concerns have been addressed. I have also read the other reviewers' questions and discussions. Overall, I am inclined to accept this paper and hope that the authors will incorporate these discussions, especially the additional experiments, into the revised version. I will increase my rating to "borderline accept".

**Limitations:**

Yes.

**Paper Formatting Concerns:**

None.

**Quality:**

3

**Strengths And Weaknesses:**

Strengths
1. The analysis of conflicting between mask ratio and correlation of noise is reasonable.
2. The proposed pixel refilling strategy could improve the tradeoff between mask ratio and correlation of noise, which is practical to existing blind-spot denoising methods.
3. The paper is well written and organized.

Weaknesses
1. The experiments on DND and PolyU dataset for real-world denoising are lacked.
2. The DBSN network architecture used in this paper may be computation inefficient, what about use U-Net based BSN? How does the size of the network's parameters affect the results?
3. There are lack of recent related works on self-supervised image denoising such as ZS-N2M[1], SASL[2], AT-BSN[3], TBSN[4], Pixel2Pixel[5]. In particular, adding more methods and results comparison with Pixel2Pixel and ZS-N2M would be better.
4. There are lack of computation comparison in the paper, please compare the training time, inference time, parameter count, etc. with previous zero-shot BSD methods.

[1]Zero-Shot Noise2Mean: Gap Minimization for Efficient Denoising from a Single Noisy Image, AAAI 2025.
[2]Spatially adaptive self-supervised learning for real-world image denoising, CVPR 2023.
[3]Exploring Efficient Asymmetric Blind-Spots for Self-Supervised Denoising in Real-World Scenarios, CVPR 2024.
[4]Rethinking Transformer-Based Blind-Spot Network for Self-Supervised Image Denoising. AAAI 2025.
[5]Pixel2Pixel: A Pixelwise Approach for Zero-Shot Single Image Denoising. TPAMI 2025.

---

> ### Author Rebuttal · Authors · 2025-07-30
>
> Thank you for the detailed and constructive comments. The main weaknesses raised concern missing experiments. which will be included in the revision.  Please see below for the detailed response regarding the evaluations (DND, NIND, PolyU), recent method comparisons, and computational analysis.
>
> ---
>
> **[W1]** *The experiments on DND and PolyU dataset for real-world denoising are lacked.*
>
> See the table below for the requested comparisons. Results on PolyU are quoted from the Pixel2Pixel paper. Due to time constraints, we conducted evaluations on the DND dataset and two NIND subsets using the official code of only competing methods, where NIND[1] is a real-world dataset consisting of clean-noisy image pairs captured at ISO levels of 3200, 4000, 5000, and 6400, with 22, 14, 13, and 79 pairs, respectively. We selected two subsets, ISO3200 and ISO5000 for comparison.
>
>  It can be seen that our method noticeably outperformed the competing methods.
>
>
> *Table C.1.1. Comparison of zero-shot denoising methods on different real denoising datasets.*
>
> |Method|DND|PolyU|NIND ISO3200|NIND ISO5000|SIDD Val|SIDD Ben|
> |-|-|-|-|-|-|-|
> |ZS-N2N| 30.67/0.737| 35.17/--|29.40/0.642|26.58/0.546|25.59/0.565|30.19/0.428|
> |Pixel2Pixel|34.16/0.836|*36.11*/--|32.30/0.783|30.18/0.745|30.29/--|31.87/0.631|
> |ScoreDVI|*36.19/0.929*|34.00/--|34.12/0.845|31.64/0.803|34.75/0.857|*35.39/0.859*|
> |MASH|34.75/0.910|31.97/--|*33.32/0.834*|*31.34/0.802*|*35.06/0.851*|34.80/0.814|
> |Ours|**36.79/0.924**|**36.29/0.942**|**34.40/0.860**|**32.58/0.835**|**36.32/0.874**|**36.81/0.874**|
>
>
> [1] *Brummer et al. (2019). Natural image noise dataset. (CVPRW).*
>
> ---
>
> **[W2]** *The DBSN network architecture used in this paper may be computation inefficient, what about use U-Net based BSN? How does the size of the network's parameters affect the results?*
>
> For the model size of our method, please refer to Appendix A of the supplemental materials for the details. Indeed, we use the small DBSN with [64, 4] channels and blocks as our final model, which has only 0.5M parameters.
>
> We apply the proposed pixel refilling scheme to a vanilla U-Net-based BSD model. Please see Table below for the results on SIDD Validation set. It shows that our method also works for the vanilla U-Net based BSD, with 1.03 dB performance gain.
>
>
>
> *Table C.2.1. The Proposed Pixel refilling on U-Net based BSD.*
>
> | Vanilla U-Net| U-Net with **Ours** |
> |---------|----------|
> | 32.72   | **33.75** |
>
>
> Regarding the performance impact of model size, please see the table below for the ablation study on model size and computational efficiency. While larger models slightly improve PSNR (e.g., +0.09 dB from 0.5M to 3.33M parameters), the increase in FLOPs is substantial (32.97G to 217.74G). To balance performance and complexity, we adopt the compact 0.5M model.
>
>
> *Table C.2.2. Ablation on model size and computational efficiency*
>
> | PSNR (dB) | Model Size [Channels, Blocks] | #Params (M) | FLOPs (G) | Runtime (s) |
> |-----------|-------------------------------|-------------|-----------|-----------|
> | 36.56     | [64, 4]                        | 0.50        | 32.97     |65.4      |
> | 36.55     | [64, 8]                        | 0.83        | 54.44     |110.2     |
> | 36.58     | [128, 4]                       | 2.02        | 131.84    |253.6     |
> | 36.65     | [128, 8]                       | 3.33        | 217.74    |420.1     |
>
> ---
>
> **[W3]** *There are lack of recent related works,  such as ZS-N2M[1], SASL[2], AT-BSN[3], TBSN[4], Pixel2Pixel[5]. In particular, Pixel2Pixel and ZS-N2M*
>
> We thank the reviewer for highlighting these recent works. We will cite them appropriately in the revised manuscript.
>
> For Pixel2Pixel, please refer to Table C.1.1 for the comparison. The results of Pixel2Pixel on SIDD Validation and PolyU are quoted from the paper and others are generated using the author's code.  It can be seen that our method noticeably outperformed Pixel2Pixel.
>
> For ZS-N2M, the authors have not released their code (their GitHub states it will be available soon), and we have not received a response to our request. Moreover, the paper reports results only on randomly selected subsets of benchmark datasets, rather than on the full benchmarks. Therefore, a fair and complete comparison to ZS-N2M  is currently not feasible at current stage.
>
>
> ---
>
> **[W4]** *Lack of computation comparison in the paper.*
>
> We thank the reviewer for highlighting this point. Please refer to the table below for the requested comparison. Our method achieves a favorable balance between performance and computational efficiency among zero-shot denoising methods. While ZS-N2N and Pixel2Pixel are faster, they suffer from a significant drop in performance. On the other hand, ScoreDVI and MASH have performance closer to ours but with comparable computational complexity.
>
>
> *Table C.4.1. Complexity comparison of zero-shot methods.*
>
> | Method                          | Infer. time (s) | Params (M) | FLOPs (G) |
> |--------------------------------|------------------|------------|-----------|
> | DIP                            | 146.2            | 13.4       | 31.06     |
> | PD-denoising                   | 0.1              | 0.7        | 46.94     |
> | ZS-N2N                         | 9.8              | 0.02        | 1.45    |
> | Pixel2Pixel                    | 24.3               | 0.2        | 2.72     |
> | Self2Self                      | 3546.5           | 1.0        | 9.55      |
> | ScoreDVI                       | 81.2             | 13.5       | 37.87     |
> | MASH                           | 75.3             | 1.0        | 11.44     |
> | **Ours**                       | 65.4        | 0.5    | 32.97 |
>
> ---
>
> Please let us know if you have any further questions. Thank you again for the helpful feedback.

---

> > ### Comment · Reviewer_N84f · 2025-08-05
> > **Thanks**
> >
> > Thank you for your response. My concerns have been addressed. I have also read the other reviewers' questions and discussions. Overall, I am inclined to accept this paper and hope that the authors will incorporate these discussions, especially the additional experiments, into the revised version. I will increase my rating to "borderline accept".

---

> > > ### Author Response · Authors · 2025-08-06
> > > **Response With Thanks**
> > >
> > > Thank you for your follow-up and for taking the time to read our responses. We are grateful that your concerns have been addressed and appreciate your engagement with the broader discussion. We will ensure that the points raised across all reviews, including the additional experiments and clarifications, are thoroughly incorporated into the revised version. Thank you again for your constructive feedback.

---

### Official Review · Reviewer_esC5 · 2025-07-02

**Clarity:** 3
**Significance:** 3
**Originality:** 3
**Rating:** 3
**Confidence:** 3

**Summary:**

This paper presents a quantitative Bayesian-risk analysis of a linear blind spot denoising model, and reveal a trade-off between noise decorrelation and pixel similarity. It shows that replacing masked pixels with those with low noise correlation can effectively reduce prediction risk. Built on the insight, a novel pixel refilling strategy is proposed, and extensive experiments on both real and synthetic datasets demonstrate its effectiveness.

**Questions:**

Questions:

1. What is the runtime and model complexity?
2. There is no ablation regarding network structure. I wonder how does the network structure affects the denoising performance? Is it possible that other methods could still achieve similar performance when using your NN architecture?

**Ethical Concerns:**

["NO or VERY MINOR ethics concerns only"]

**Final Justification:**

I would like to lower my rating, based on the following reasons:
(1) Missing important ablation: The authors do not provide comparison on the same network structure, which is an important experiment that could show the real potential of the proposed methodology. Also, the updated FLOPs information show that the adopted network takes more FLOPs and has double inference time compared to MASH. It implies that the achieved SoTA performance could partly attribute to the bigger network.
(2) I also refer to the comments from other reviewers, and their concerns have not been well addressed yet.

**Limitations:**

yes

**Paper Formatting Concerns:**

No paper formatting concerns.

**Quality:**

3

**Strengths And Weaknesses:**

Strengths:

1. The idea of pixel refilling is interesting and novel. The idea also has solid theoretical analysis and support.
2. The proposed method achieves state-of-the-art denoising performance among the zero-shot denoising approaches with a clear margin on both synthetic and real benchmark datasets.
3. The paper is well organized. The presentation of the idea, methodology, and experimental details are clear. Also, the supplementary material supplements enough technical details for the readers.

Weaknesses:

1. It could be better if the runtime and parameter complexity can be provided and compared.

---

> ### Author Rebuttal · Authors · 2025-07-30
>
> We thank the reviewer for the encouraging comments and constructive feedback. We are  glad that the novelty of our pixel refilling strategy and the strength of our theoretical analysis were recognized. Please see below for the responses to the concerns.
>
> ---
> **[W1&Q1]** *What is the runtime and model complexity?*
>
>
> Please refer to Table B.1.1. below for the comparison of inference time, parameter count, and FLOPs.. Our method achieves a favorable balance between performance and computational efficiency among zero-shot denoising methods. While ZS-N2N and Pixel2Pixel are faster, they suffer from a significant drop in performance. On the other hand, ScoreDVI and MASH have performance closer to ours but with comparable computational complexity.
>
> *Table B.1.1. Complexity comparison of zero-shot methods.*
>
> | Method                          | Infer. time (s) | Params (M) | FLOPs (G) |
> |--------------------------------|------------------|------------|-----------|
> | DIP                            | 146.2            | 13.4       | 31.06     |
> | PD-denoising                   | 0.1              | 0.7        | 46.94     |
> | ZS-N2N                         | 9.8              | 0.02        | 1.45    |
> | Pixel2Pixel                    | 24.3               | 0.2        | 2.72     |
> | Self2Self                      | 3546.5           | 1.0        | 9.55      |
> | ScoreDVI                       | 81.2             | 13.5       | 37.87     |
> | MASH                           | 75.3             | 1.0        | 11.44     |
> | **Ours**                       | 65.4        | 0.5    | 32.97 |
>
>
>
>
> ---
>
> **[Q2]** *How does the network structure affects the denoising performance? Is it possible that other methods could still achieve similar performance when using your NN architecture?*
>
> We thank the reviewer for highlighting the need for ablation studies on network structure. In response, we conducted a model ablation on the SIDD Validation set (200 randomly selected images due to time constraints), analyzing model size and computational cost using a DBSN-based architecture parameterized by the number of channels and blocks. Details of the architecture are provided in Appendix A of the supplemental material. As shown in Table B.2.1, we adopt the compact [64, 4] model in our final version to strike a balance between performance and efficiency.
>
> *Table B.2.1. Ablation on model size and computation comparisons*
>
> | PSNR (dB) | Model Size [Channels, Blocks] | #Params (M) | FLOPs (G) | Runtime (s) |
> |-----------|-------------------------------|-------------|-----------|-----------|
> | 36.56     | [64, 4]                        | 0.50        | 32.97     |65.4      |
> | 36.55     | [64, 8]                        | 0.83        | 54.44     |110.2     |
> | 36.58     | [128, 4]                       | 2.02        | 131.84    |253.6     |
> | 36.65     | [128, 8]                       | 3.33        | 217.74    |420.1     |
>
> We also apply our method to a vanilla U-Net-based BSD model on the SIDD Validation set. Results are shown in Table B.2.2. It shows that our method can consistently enhance the performance of vanilla U-Net based BSD by 1.03 dB.
>
> *Table B.2.2. Pixel refilling on U-Net based BSD.*
>
> | Vanilla U-Net | U-Net with **Ours** |
> |---------|----------|
> | 32.72   | **33.75** |
>
> ---
>
> We will include these results in the revised manuscript. Thank you again for your positive comments and valuable suggestions.

---

> ### Comment · Reviewer_esC5 · 2025-08-03
> **Response to the author rebuttal**
>
> 1. I noticed that the runtime data comes from the paper of MASH. However since the type of GPU was not given in MASH paper, it would be more fair if you test the inference time of all compared methods at your own GPU.
> 2. Regarding the network structure, to prove the proposed method is more effective than the previous ones, I think you should use the same network structure as MASH and Noise2Noise.

---

> > ### Author Response · Authors · 2025-08-04
> >
> > **[Q1]** *Runtime issue*
> >
> > *Table B.1.1. Complexity comparison of zero-shot methods on A6000*
> >
> > | Method |          PSNR               | Infer. time (s) | Params (M) | FLOPs (G) |
> > |---------|-----------------------|------------------|------------|-----------|
> > | DIP      |        32.11/0.740              | 89.0            | 13.4       | 31.06     |
> > | PD-denoising|        33.97/0.820            | 0.1              | 0.7        | 46.94     |
> > | ZS-N2N       |     25.59/0.565              | 12.5              | 0.02        | 1.45    |
> > | Pixel2Pixel   |       30.29/--            | 15.8               | 0.2        | 2.72     |
> > | Self2Self      |      29.46/0.595          | 1273.4           | 1.0        | 9.55      |
> > | ScoreDVI        |       34.75/0.856          | 66.2             | 13.5       | 37.87     |
> > | MASH             |      35.06/0.851         | 39.2             | 1.0        | 11.44     |
> > | **Ours(1k iters)**         |  36.08/0.874              | 31.1       | 0.5    | 32.97 |
> > | **Ours(final)**         |   36.32/0.875             | 65.4        | 0.5    | 32.97 |
> >
> > Thank you for highlighting the runtime issue. Due to the large number of additional experiments conducted during the rebuttal period and limited GPU availability, the runtime table was compiled progressively. In prioritizing baselines whose reported results were obtained on hardware significantly different from ours (A6000), some inconsistencies, such as MASH was likely evaluated on a comparable GPU (e.g., 3090), were inadvertently overlooked.
> >
> >
> > Since no additional  experiments were requested by now, we have now carefully re-evaluated the runtime of all methods under a consistent hardware setting (NVIDIA A6000 GPU). We report both early-stage (1k-iteration) and final results to illustrate the convergence behavior of our method. It can be seen that our method already surpasses MASH on the SIDD validation set after 1000 iterations,  in terms of PSNR/SSIM (36.08/0.874 vs. 35.06/0.851) and runtime (31.1s vs. 39.2s). The final results (2k-iteration) yield improved performance at the cost of additional runtime (36.32/0.875 in 65.4s).
> >
> > **[Q2]** *Effectiveness of our proposed method*
> >
> > Thank you for your insightful suggestion. Right now, due to time constraints, we are currently unable to provide results incorporating our pixel refilling strategy into the MASH network,  but we will include these experiments in the revised manuscript.

---

### Official Review · Reviewer_gdnC · 2025-07-03

**Clarity:** 3
**Significance:** 2
**Originality:** 2
**Rating:** 4
**Confidence:** 4

**Summary:**

The paper proposes a self-supervised image denoising method applied at test time (test-time training). The approach enhances the vanilla Blind-Spot Denoising (BSD) framework by refilling masked pixels with pixels from similar patches within the same image, or its downsampled version. The method achieves competitive results compared to other self-supervised single-image denoising methods.

**Questions:**

See weaknesses.

**Ethical Concerns:**

["NO or VERY MINOR ethics concerns only"]

**Final Justification:**

The paper achieves state-of-the-art results across multiple datasets with a consistent hyperparameter configuration, as claimed by the authors, and introduces a novel pixel-refilling approach for blind-spot denoising. However, an ablation study on the chosen network/backbone is necessary to determine whether pixel refilling and pixel-shuffle downsampling alone are sufficient to achieve these results (if not, the architecture itself should be highlighted as a third component of the method). A more thorough robustness analysis of the pixel-shuffling idea in various scenarios (e.g., high noise levels, textured regions) is also needed. If a heavy architecture is crucial to the state-of-the-art performance, the authors should acknowledge that the improvement comes at the cost of increased computational requirements

Overall, I have a mixed opinion and lean toward borderline acceptance, based on the authors’ efforts during the discussion, and I recommend incorporating the above suggestions in their revision.

**Limitations:**

See weaknesses.

**Paper Formatting Concerns:**

None.

**Quality:**

2

**Strengths And Weaknesses:**

**Strengths:**

1. The paper tackles the interesting problem of unsupervised image denoising.

2. Although the theoretical analysis is based on simplistic assumptions and does not seem to lead to novel findings, I found it overall insightful.

3. The method shows good improvement over the baselines on the SSID dataset.


**Weaknesses:**

1. The central idea and claimed contribution focus on improving the BSD framework. However, the authors train their model using a combination of the BSD loss and the pixel shuffle downsampling loss with $\lambda=0.1$. The contribution of each component should be investigated, particularly the case of $\lambda=0$, which would clearly highlight the contribution of the proposed method.

2. The only significant improvement over the baselines in case of the real data is observed on the SIDD dataset, which is generally characterized by (i) a high noise level and (ii) high noise correlation. I suspect that the noise level is an important factor in the success of the method, and I believe investigating this aspect is crucial. In particular, I think that the proposed pixel refilling strategy might be suboptimal in cases of low noise levels, and that a vanilla BSD with a low masking ratio could lead to better results.

---

> ### Author Rebuttal · Authors · 2025-07-30
>
> Thank you for the feedback. We emphasize that our method introduces a novel pixel refilling strategy for blind-spot denoising (BSD), grounded in our solid theoretical framework. Unlike typical zero-shot BSD approaches, e.g., the current SOTA zero-shot method MASH [1], which rely solely on zero-masking, our approach replaces the masked pixels with meaningful candidates from the same image or its downsampled version. While it incorporates standard techniques such as pixel shuffle downsampling (commonly asymmetrically used due to its limitations in BSD variants like APBSN[2], LGBPN[3] and PUCA[4], as we have discussed in Sec. 2 of our manuscript), the main contribution lies in the refilling strategy, which not only drives significant performance gains but also addresses the limitations of shuffling. In response to your comments, we have conducted ablation studies and additional experiments, which will be included in the revised manuscript. Please see our detailed responses below.
>
> [1] *Chihaoui, H., et al. (2024). Masked and shuffled blind spot denoising for real-world images. CVPR.*
>
> [2] *Lee et al. (2022). AP-BSN: Self-supervised denoising for real-world images via asymmetric PD and blind-spot network. CVPR.*
>
> [3] *Wang et al. (2023). LG-BPN: Local and global blind-patch network for self-supervised real-world denoising. CVPR.*
>
> [4] *Jang et al. (2023). PUCA: Patch-unshuffle and channel attention for enhanced self-supervised image denoising. NeurIPS.*
>
> ---
>
> **[W1]** *The authors train their model using a combination of the BSD loss and the pixel shuffle downsampling loss with $\lambda = 0.1$. The contribution of each component should be investigated, particularly the case of $\lambda = 0$*
>
> Thanks for the insightful suggestion. Please refer to Table A.1.1 below for the ablation study. We validate the contribution of pixel refilling on two real-world datasets, SIDD validation and NIND[5] subset, where NIND is a real-world dataset consisting of clean-noisy image pairs captured at ISO levels of 3200, 4000, 5000, and 6400, with 22, 14, 13, and 79 pairs, respectively and we pick up the smallest subset.
>
> Pixel shuffle downsampling alone slightly degrades performance ($34.12\rightarrow 34.01$ on SIDD and $29.90\rightarrow 29.83$ on NIND subset), as it breaks local noise correlation but also weakens pixel correlation. In contrast, our proposed pixel refilling significantly improves performance ($34.12\rightarrow 34.95$ on SIDD and $29.90\rightarrow 32.10$ on NIND subset) by introducing distant pixels with strong pixel correlation and low noise correlation. When combined, the two strategies complement each other: pixel reshuffle helps decorrelate local noise, while pixel refilling restores meaningful pixel correlations without compromising noise independence. Together, they yield the best performance ($35.65$ on SIDD and $32.58$ on NIND subset). These empirical observations are consistent with our theoretical analysis.
>
>
> *Table A.1.1. Ablation study of two components.*
> | PD Loss |  Pixel Refilling in BSD|SIDD (PSNR dB) | NIND (PSNR dB)|
> |-|-|-|-|
> | $\times$|$\times$|34.12 (Baseline)| 29.90(Baseline)|
> | $\times$|$\checkmark$|34.95 ($\uparrow$ 0.83)|32.10 ($\uparrow$ 2.20)|
> | $\checkmark$|$\times$|34.01 ($\downarrow$ 0.11)|29.83 ($\downarrow$ 0.07)|
> | $\checkmark$|$\checkmark$|35.65 ($\uparrow$ 1.53)|32.58 ($\uparrow$ 2.68)|
>
>
> ---
>
>
> **[W2]** *The only significant improvement over the baselines in case of the real data is observed on the SIDD dataset. The proposed method might be sub-optimal in cases of low noise levels, and that a vanilla BSD with a low masking ratio could lead to better results.*
>
> We thank the reviewer for this insightful comment. In response, we conducted the experiments under various noise conditions. Please see Table A.2.1. for the results across multiple real-world datasets, including DND[6], PolyU[7], NIND[5] subsets ISO3200, and ISO5000. DND consists of 50 noisy-clean pairs, formed by shooting the same scene twice with different ISO values. PolyU contains 40 pairs of noisy-clean images captured at various ISO levels. These datasets span a wide range of noise levels. In particular, on the PolyU dataset, which features the lowest noise level, our method still achieves the best performance. This suggests that the proposed pixel refilling strategy remains effective even when the noise level is low.
>
> *Table A.2.1. Comparison of zero-shot denoising methods on different real denoising datasets.*
>
> |Method|DND|PolyU|NIND ISO3200|NIND ISO5000|SIDD Val|SIDD Ben|
> |-|-|-|-|-|-|-|
> |ZS-N2N| 30.67/0.737| 35.17/--|29.40/0.642|26.58/0.546|25.59/0.565|30.19/0.428|
> |Pixel2Pixel|34.16/0.836|*36.11*/--|32.30/0.783|30.18/0.745|30.29/--|31.87/0.631|
> |ScoreDVI|*36.19/0.929*|34.00/--|34.12/0.845|31.64/0.803|34.75/0.857|*35.39/0.859*|
> |MASH|34.75/0.910|31.97/--|*33.32/0.834*|*31.34/0.802*|*35.06/0.851*|34.80/0.814|
> |Ours|**36.79/0.924**|**36.29/0.942**|**34.40/0.860**|**32.58/0.835**|**36.32/0.874**|**36.81/0.874**|
>
>
> To further examine our method's performance under low noise levels, we conduct experiments on the Kodak dataset with AWGN $\sigma=10$ and $\sigma=25$. We compare our method against vanilla BSD with low (0.1) and moderate (0.5) mask ratios to evaluate its effectiveness without relying on high noise levels or strong noise correlation. The results, presented in Tables A.2.2 and A.2.3, show that our method consistently outperforms the vanilla BSD baseline across both noise settings.
>
> *Table A.2.2. Comparison on Kodak with $\sigma = 10$ (low noise).*
>
> | Method                         | Mask Ratio = 0.5 | Mask Ratio = 0.1 |
> |--------------------------------|------------------|------------------|
> | Vanilla BSD                    | 29.75            | 30.96            |
> | Ours                           | *30.98*          | **31.39**        |
>
> *Table A.2.3. Comparison on Kodak with $\sigma = 25$ (moderate noise).*
>
> | Method                         | Mask Ratio = 0.5 | Mask Ratio = 0.1 |
> |--------------------------------|------------------|------------------|
> | Vanilla BSD                    | 27.91            | 28.77            |
> | Ours                           | *29.03*          | **29.12**        |
>
>
>
> [5] *Brummer et al. (2019). Natural image noise dataset. (CVPRW).*
>
> [6] *Plotz et al. (2017). Benchmarking denoising algorithms with real photographs. (CVPR).*
>
> [7] *Xu et al. (2018). Real-world noisy image denoising: A new benchmark. (arXiv).*
>
> ---
>
> We welcome any further questions and thank the reviewer again for the valuable and constructive feedback.

---

> ### Comment · Reviewer_gdnC · 2025-08-04
>
> Thanks for the your rebuttal. Could you clarify the following points:
>
> 1) Regarding the ablation study in Table A.1.1: Is it conducted on the full SIDD dataset? A standard BSD baseline typically performs poorly on SIDD due to its correlated noise characteristics. However, the baseline performance reported (PSNR of 34.12) appears quite strong compared to other methods. Could you elaborate on this?
> Specifically, APBSN-single achieves a PSNR of 30.90 on SIDD, and your baseline seems to use a similar network. How do you explain this performance gap?
>
> 2) On the results in Table A.2.1: are all results obtained using the same set of hyperparameters?
> Could you mention the used hyper-parameters, in particular the masking ratio?
>
> 3) In the case of highly noisy images, do you believe that relying on patch similarity is sufficiently robust? Does the refilling in this scenario become simply a random refilling? I’m particularly concerned that the refilling strategy may not be beneficial under high i.i.d. noise conditions, where similar patches may be difficult to identify accurately.
>
> 4) I think the FMDD dataset includes a diverse set of noise patterns, ranging from low to high noise levels and from i.i.d. to correlated noise. These variations can pose unique challenges for denoising methods. Could you comment on why your method does not achieve the best performance in this setting?
>
> 5) While the theoretical study presented in Section 3.1 is interesting, I find its initial assumptions somewhat restrictive, and not necessarily mean that the method should work well in real-world. In particular, pixels refilling might not always be optimal, which could potentially negate the advantages claimed by the method.
>  It would be valuable to include an analysis of where the idea of pixel refilling tends to fail or underperform. For example, a controlled comparison that varies one factor at a time, such as correlated vs. i.i.d. noise, low vs. high noise levels, and textured vs. flat image regions, would help to better understand the robustness and limitations of the proposed idea.

---

> ### Author Response · Authors · 2025-08-05
>
> We appreciate the reviewer’s insightful follow-up and the opportunity to elaborate on the key aspects of our work. Please see our point-by-point clarifications below.
>
> **[Q1]**
>
> Yes, the results in Table A.1.1 are obtained on the full SIDD validation dataset. The relatively strong baseline performance of BSN can be attributed to its implementation, which replaces standard convolutions with dilated convolutions. This expands the receptive field and improves the model's  ability to handle the spatially correlated noise characteristic of SIDD. In contrast, the weaker performance of APBSN-single (PSNR 30.90), which is also implemented with dilated convolutions, is primarily due to its asymmetric pixel-shuffling downsampling (PD) stride configuration, using stride 5 during training and stride 2 during inference. This mismatch leads to over-fitting in the single-image setting, where the model is trained and evaluated on the same image but with structurally inconsistent inputs due to differing down-sampling strategies. When trained on a dataset, the model benefits from a variety of image structures, which helps mitigate the negative effects of the stride mismatch, although some residual impact remains.
>
> **[Q2]**
>
> Yes, they all use the same hyperparameters by abaltion from SIDD Validation, mask ratio to be 0.3 and $\lambda=0.2$.
>
> **[Q3]**
>
>
> Yes. We agree that as the noise level increases, patch matches become less reliable and the matched pixels exhibit weaker intensity correlation. Nevertheless, as shown in Table 2 of our manuscript, for AWGN with $\sigma=50$ (noisy observation has PSNR $\approx$ 14 dB on Kodak), our method still performs well, achieving an average PSNR of 26.58 dB, about 2 dB higher than MASH. To assess the limit under extreme noise, we conducted additional experiments on Kodak corrupted by AWGN with $\sigma=100$ (noisy data has PSNR $\approx$ 8dB). The results below indicate that refilling via patch matching provides only a marginal benefit.
>
> *Table 3.1: Performance Comparison on Kodak with AWGN $\sigma=100$*
> |Method|PSNR |
> |-|-|
> |Baseline|23.26|
> |With Refilling|23.36|
>
> In summary, the proposed approach is affected by noise level: it remains beneficial under high noise level and becomes less effective as noise approaches extreme levels.
>
>
> **[Q4]**
>
> Thank you for the question. The main challenge on FMDD is less about diverse noise regimes and more about content: images of biological particles exhibit weaker patch self-recurrence than natural images. Consequently, patch matching yields varying refill quality, in some regions the matches are informative, while in others they are unreliable. Ideally, different quality of pixel refilling requires different mask ratio, as shown in our analysis in Appendix C. To be fair, we use a constant masking ratio for all images to maintain consistency across FMDD, which explains why it does not achieve the best overall performance. In future work, we plan to develop an adaptive masking strategy that adjusts the mask ratio based on image content (e.g., quality of patch matching), which we expect will improve performance on images with weak self-recurrence of local structures.
>
>
> **[Q5]**
>
>
> We agree that the assumptions in Sec. 3.1 provide a simplified model and, by themselves, do not guarantee strong real-world performance, this limitation is common, as an exact model of real-world noise remains open. In practice, the benefit of pixel refilling via patch matching is most evident when: (i) the image exhibits sufficiently strong local self-recurrence (so patch matches are reliable), and (ii) the noise level is not extreme (so matches retain intensity correlation). When either condition is violated (weak self-recurrence or very high noise), the benefit of refilling can become neutral or marginal.
>
> To clarify applicability and limitations, we will add a comprehensive, factorized evaluation in the revision that varies one axis at a time: noise type, noise levels, and content. We will also check key configuration choices, masking ratio, patch size, search radius, and a confidence gate for matches. This study will identify the regimes where refilling is robust and where it underperforms. In future, we will study prototype adaptive masking (content- and confidence-aware) as a potential improvement for images with weak self-recurrence.

---

> ### Comment · Reviewer_gdnC · 2025-08-09
>
> Thank you for your response.
>
> To me, the strongest point of the paper is that it achieves new state-of-the-art results across multiple benchmarks (SIDD, DND,..) using the same hyperparameters (masking ratio, $\lambda$, number of iterations), as claimed by the authors.
>
> My main three criticisms of the current version:
>
> 1. I have concerns that a vanilla BSD with the standard U-Net architecture used in prior work, without any tricks, could even reach a PSNR of 33 dB on SIDD for example. With the additional gain from pixel refilling and pixel shuffle downsampling, I am not certain the method could truly reach state-of-the-art performance. So, an ablation study about the used architecture is necessary. If the architecture plays a decisive role in achieving state-of-the-art results, it would be misleading to attribute the performance solely to the mentioned two components and  In that case, the method should highlights the three components: i) pixels refilling, ii) pixel shuffle downsampling, iii) a better architecture choice for test-time denoising that addresses the weaknesses of APBSN-single.
>
> 2. Generally, any idea has its limitations, and pixel refilling is no exception. Since it is the central idea of the paper, the limitations section should include some aspects about it. In this context, an analysis of scenarios where pixel refilling tends to fail or underperform would be valuable. (The authors have committed to adding this in their revision so it should be fine)
>
> 3. The computational comparison should be included in the main paper. If the method relies on a heavy network to achieve state-of-the-art results, the authors should acknowledge that the improved performance comes at the cost of increased computational requirements.
>
> Overall, I believe I have reasons to hold both views. Given the effort the authors put into the rebuttal and discussion, I lean towards a positive assessment. I hope they address my remaining concerns and openly highlight their paper’s limitations, such as limitations of pixel refilling, reliance on a specific architecture, and compute budget, instead of hiding them, as this transparency will benefit future work and ultimately increase the value of their contribution.

---

> > ### Author Response · Authors · 2025-08-09
> > **Response With Thanks**
> >
> > We sincerely thank the reviewer for the insightful discussion and the positive assessment, which have greatly helped us refine and improve the paper. As discussed throughout the rebuttal, we will strengthen the final version by adding a clear ablation study to isolate the contributions of pixel refilling, pixel shuffle downsampling, and the architecture choice, including a dedicated limitations section, providing a detailed computational comparison with other methods, and presenting the trade-off between performance gains and increased computation. We appreciate your engagement in the discussion and will ensure these points are clearly and openly presented in the final version.

---

### Comment · Area_Chair_tVCY · 2025-08-03
**Reviewers please respond to the rebuttal!**

Dear reviewers,

if you have not yet responded to the rebuttal of the authors, please do so as soon as possible, since the rebuttal window closes soon.

Please check whether all your concerns have been addressed!  If yes, please consider raising your score.

Best wishes,
your AC

---

### Decision · Program_Chairs · 2025-09-17

**Decision:**

Accept (poster)

**Comment:**

Another contribution to the blind-spot denoising literature.  In this case, correlated noise is considered and an interesting trick of refilling pixels is presented.  The paper is well written and the results are good.  However, some additional ablation could clarify the method further.    Some aspects might remain unclear, such as the influence of the size of the used architecture.  Overall the reviewers are leaning to accept to I suggest a poster-accept with somewhat low confidence.